# Hamiltonian structure of 2D fluid dynamics with broken parity

Gustavo Machado Monteiro[1*], Alexander G. Abanov[2,3] and Sriram Ganeshan[1,4]

**1** Department of Physics, City College, City University of New York,
New York, NY 10031, USA
**2** Simons Center for Geometry and Physics, Stony Brook, NY 11794, USA
**3** Department of Physics and Astronomy, Stony Brook University,
Stony Brook, NY 11794, USA
**4** CUNY Graduate Center, New York, NY 10031, USA

⋆ gmachadomonteiro@ccny.cuny.edu

Isotropic fluids in two spatial dimensions can break parity symmetry and sustain transverse stresses which do not lead to dissipation. Corresponding transport coefficients include odd viscosity, odd torque, and odd pressure. We consider an isotropic Galilean invariant fluid dynamics in the adiabatic regime with momentum and particle density conservation. We find conditions on transport coefficients that correspond to dissipationless and separately to Hamiltonian fluid dynamics. The restriction on the transport coefficients will help identify what kind of hydrodynamics can be obtained by coarse-graining a microscopic Hamiltonian system. Interestingly, not all parity-breaking transport coefficients lead to energy conservation and, generally, the fluid dynamics is energy conserving but not Hamiltonian. We show how this dynamics can be realized by imposing a nonholonomic constraint on the Hamiltonian system.

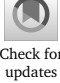

# 1  Introduction

In fluid dynamics, viscosities appear as transport coefficients in the first-order derivative expansion of the stress tensor. Viscosity terms preserve both mass and momentum conservation laws but usually spoil the energy conservation due to their dissipative nature. For example, shear viscosity ($\eta$) introduces friction between adjacent fluid layers that do not flow with the same velocity, whereas the bulk viscosity ($\zeta$) provides resistance to compression or expansion of the fluid.

In two spatial dimensions, there exist viscosity coefficients within the first-order hydrodynamics that break parity symmetry and preserve both the fluid isotropy and energy conservation. Odd viscosity ($\eta_H$) is undoubtedly the most famous of the parity-breaking viscosity terms, first showing up in the study of plasma physics [1] and later on as a new quantized response in quantum Hall systems [2]. It was introduced in the hydrodynamic context by Avron in [3] and was recently experimentally observed in both electron fluids [4] and active matter systems [5].

In quantum Hall systems, the odd viscosity is associated with the intrinsic angular momentum density of the electron fluid [6–8]. In classical systems, the intrinsic angular momentum density ($\ell$) is an independent dynamical variable with its own continuity equation. On the other hand, odd viscosity is a transport coefficient, that is, a function of density and temperature. If we initialize $\ell$ to be proportional to density, this relationship is preserved for all times since both quantities satisfy similar continuity equations (in the absence of internal torque). This class of initial conditions with $\ell \propto \rho$ corresponds to a projected Hamiltonian system. In general, such a projection need not lead to a new Hamiltonian system. However, $\ell \propto \rho$ does not spoil the underlying Poisson algebra as shown in Ref. [9]. Together with a velocity redefinition [8], this projection gives rise to odd viscosity terms in the momentum conservation equation. Physically, this projection ($\ell \propto \rho$) can be realized in systems where fluid intrinsic angular momentum equilibrates much faster than the other hydrodynamic quantities [10].

Recently, it was shown in Ref. [11] that odd viscosity could also arise in the equation of motion from a non-Hamiltonian reduction of the intrinsic angular momentum $\ell$. For that, the authors introduced dissipative terms and an external drive to a Hamiltonian system. The out-of-equilibrium dynamics leads to the relaxation of the fluid intrinsic angular momentum, giving rise to the odd viscosity term in the Navier-Stokes equation. The non-Hamiltonian projection in Ref. [11], odd viscosity is a linear function of the mass density, but it depends explicitly on the external drive. Although Refs. [9–11] describe different physical systems, all of them obtain odd viscosity through relaxation of the intrinsic angular momentum. However, odd viscosity is not the only parity odd coefficient in two dimensions and it is not *a priori* clear if some or all of these other coefficients can be obtained starting from a microscopic Hamiltonian system. In fact, the identification of Hamiltonian systems provides a benchmark for idealized dissipationless systems about which the dissipative contributions can be introduced.

In this work, we start from the most general first-order hydrodynamic equations of motion and derive under which conditions the density dependent viscosity coefficients can be derived from a Hamiltonian system. Throughout the paper, we only consider adiabatic flows, neglect-

ing heat transport.[1] There are in total 6 independent viscosity coefficients which preserve fluid isotropy and satisfy Galilean symmetry in two dimensions. Half of these transport coefficients are even under parity symmetry, and the other half is parity odd.[2] In general, parity-violating forces are transverse to the fluid motion and perform no work. Therefore, such terms are expected not to dissipate energy. However, it is not apparent whether a 2D hydrodynamical system with parity-odd coefficients possesses conserved energy in general. In addition, even if the conserved energy exists, it is not obvious that the corresponding system is Hamiltonian. We show that not all parity-breaking transport coefficients amount to energy conservation. For a hydrodynamic system whose energy is conserved, we derive sufficient conditions on the transport coefficients for the system to be Hamiltonian. As a consequence, we obtain that an energy-conserving hydrodynamic system is Hamiltonian if there exists a conserved quantity, $\rho v_i + \epsilon_{ij}\partial_j \eta_H$, which satisfies the diffeomorphism algebra. This quantity is associated with the "molecular" center-of-mass momentum density, as pointed out in [11]. The energy-conserving cases that fail to be Hamiltonian systems are closely related to projections of the intrinsic angular momentum incompatible with the Poisson algebra. We discuss the projection of the intrinsic angular momentum to a function of mass density and the breakdown of the Hamiltonian system from the point of view of nonholonomic constraints.

This paper is organized as follows: we begin by defining our hydrodynamic system in Sec 2 and present the conditions for the fluid energy to be conserved. In Sec. 3, we derive under which conditions the aforementioned energy-conserving systems are Hamiltonian. In Sec. 4, we study the connection between dynamical intrinsic angular momentum and odd viscosity as well as its implications towards non-Hamiltonian systems with conserved energy density. We close the paper with conclusions and discussions. Some technical details are relegated to appendices.

## 2 Energy conservation in 2D fluid dynamics

Hydrodynamic equations consist of local conservation laws for mass and momentum, assuming all other relevant quantities are equilibrated. These equations are supplemented by constitutive relations between the conserved quantities. The presence of a finite mean-free-path and a finite characteristic relaxation time of the interacting system modify the dynamics at small length scales and at transient times, giving rise to derivative corrections in these constitutive relations. This means that constitutive relations can be formally written as an expansion in derivatives, both in time and space, and the hydrodynamic equations are obtained by truncating this series at some particular order. In non-relativistic hydrodynamics, spatial and time derivatives do not scale the same way, and only terms with a single spatial derivative enter in the constitutive relations in the first-order derivative expansion. Mass (the continuity equation) and momentum conservation can be written in terms of the mass density ($\rho$) and velocity ($v_i$) as

$$\partial_t \rho + \partial_i(\rho v_i) = 0, \tag{1}$$

$$\partial_t v_j + v_i \partial_i v_j = \frac{1}{\rho}\partial_i T_{ij}. \tag{2}$$

Here, the stress tensor $T_{ij}$ is of first-order in spatial gradients and is given by

$$T_{ij} = -p(\rho)\delta_{ij} + \eta_{ijkl}(\rho)\partial_k v_l, \tag{3}$$

---

[1]Adiabatic conditions are satisfied when the fluid contracts or expands so fast that there is no time to exchange heat between its adjacent layers. The adiabaticity ensures that energy conservation follows directly from mass and momentum conservation laws.

[2]Throughout this paper, we denote parity-breaking terms with a subscript $H$.

where $\eta_{ijkl}$ is the viscosity tensor. In principle, all transport coefficients must be functions of density and temperature, however, for adiabatic flows, temperature can be expressed in terms of fluid density. The relation between pressure and mass density for adiabatic flows gives us the equation of state,

$$p(\rho) = \rho\, \varepsilon'(\rho) - \varepsilon(\rho), \tag{4}$$

where $\varepsilon(\rho)$ is the internal energy density of the fluid.

In first-order hydrodynamics, the fluid velocity cannot be uniquely defined, leading to different hydrodynamic frames [12–15]. Even though hydrodynamic equations depend on the specific parametrization of momentum density in terms of the fluid velocity, the momentum conservation must not rely on any particular definition of the fluid velocity. In this work, we *define* the fluid velocity such that the momentum density is expressed as $\rho v_i$. Equations (1, 2) must be invariant under the Galilean symmetry, that is,

$$t \to t, \quad x_i \to x_i - V_i\, t, \quad \text{and} \quad v_i \to v_i + V_i,$$

for a constant boost velocity $V_i$. Consequently, the divergence of the stress tensor must be invariant under this Galilean symmetry; that is, the force must be independent of the boost velocity. Moreover, this also implies that the mass current can only differ from the momentum density by some "magnetization current", which does not modify the equations of motion (1, 2).

The isotropic condition imposes that there are only 6 independent viscosity coefficients in two dimensions, that is,

$$\begin{aligned}
\eta_{ijkl} = {}& \eta\left(\delta_{ik}\delta_{jl} + \delta_{il}\delta_{jk} - \delta_{ij}\delta_{kl}\right) + \zeta\,\delta_{ij}\delta_{kl} + \Gamma\,\epsilon_{ij}\epsilon_{kl} \\
& + \eta_H\left(\epsilon_{ik}\delta_{jl} + \epsilon_{jl}\delta_{ik}\right) + \zeta_H\,\delta_{ij}\epsilon_{kl} + \Gamma_H\epsilon_{ij}\delta_{kl}.
\end{aligned} \tag{5}$$

Here and in the following, we suppress the dependence of all coefficients on density using the notation $\eta(\rho) \to \eta$, etc.

As previously mentioned, $\eta$, $\zeta$, and $\eta_H$ are shear, bulk, and odd viscosities, respectively. The quantity $\zeta_H$ is the odd pressure coefficient, $\Gamma$ is the rotational viscosity, and we refer to the $\Gamma_H$ term as the odd torque coefficient. Rotational viscosity gives rise to torque when the fluid vorticity is non-zero, and the odd pressure coefficient generates pressure when fluid vorticity does not vanish. Finally, the odd torque coefficient $\Gamma_H$ generates torque when the fluid expands or compresses.[3]

A close inspection of Eqs. (3) and (5) shows that there is a symmetry among transport coefficients that leaves Eq. (2) invariant. Indeed, under the transformation

$$\eta \to \eta + c_1, \qquad \zeta \to \zeta - c_1, \qquad \Gamma \to \Gamma - c_1, \tag{6}$$

$$\eta_H \to \eta_H + c_2, \quad \zeta_H \to \zeta_H - c_2, \quad \Gamma_H \to \Gamma_H + c_2, \tag{7}$$

with two arbitrary constants $c_1$ and $c_2$, we obtain

$$\partial_i T_{ij} \to \partial_i\left[T_{ij} + 2\partial_i^*\left(c_1 v_j^* + c_2 v_j\right)\right] = \partial_i T_{ij}. \tag{8}$$

Here and in the following, we define the star operation as $a_i^* \equiv \epsilon_{ij}a_j$. Since we are only interested in equations of motion and not in a particular form of the stress tensor, we will ignore these redundancies for the rest of this work. The particular form of the stress tensor, however, is crucial for the free surface problems for which a no-stress boundary condition is imposed [16].

---

[3]For a hydrodynamic system without any internal torque, the stress tensor must be symmetric, which imposes that $\Gamma = \Gamma_H = 0$.

For reasons that will be clear later, it is convenient to parametrize the parity-breaking part of Eq. (5), i.e. viscosity coefficients with subscript $H$, as

$$\eta^H_{ijkl} = \bar{\eta}_{ijkl} - \rho G' \delta_{ij} \epsilon_{kl} \, . \tag{9}$$

Here we have introduced the tensor

$$\bar{\eta}_{ijkl} = \eta_H \left( \epsilon_{ik} \delta_{jl} + \epsilon_{jl} \delta_{ik} \right) + \Gamma_H \left( \epsilon_{ij} \delta_{kl} - \delta_{ij} \epsilon_{kl} \right) \, , \tag{10}$$

which is anti-symmetric with respect to the interchange of following pairs of indices

$$\bar{\eta}_{ijkl} = -\bar{\eta}_{klij} \, . \tag{11}$$

Comparing (5) with (9,10), we see that the newly introduced function $G(\rho)$ is related to $\zeta_H$ by

$$\zeta_H = -\Gamma_H - \rho G' \, . \tag{12}$$

In order to study the Hamiltonian structure of Eqs. (1-3,5) the first step is to obtain under which conditions these equations allow for a third conserved quantity, namely, energy. We are looking for a conserved energy density $\mathscr{E}$ satisfying

$$\partial_t \mathscr{E} + \partial_i Q_i = 0 \, , \tag{13}$$

with some local energy current $Q_i$. To be consistent with zeroth-order hydrodynamics, that is, $\eta_{ijkl} \to 0$, the energy density should have the form

$$\mathscr{E} = \tfrac{1}{2} \rho v_i^2 + \varepsilon(\rho) + \dots \, , \tag{14}$$

where dots denote terms of higher order in spatial gradients of density and velocity fields. Here and in the following, we use $v_i^2$ instead of $v_i v_i$ to shorten up the notation.

We now state the general condition (up to the redundancies (6-8) in the stress tensor) for the energy conservation while leaving the full details of the calculation to the Appendix A.

**Statement I.** *The energy of a hydrodynamic system described by Eqs. (1-5) is only conserved when the parity-preserving viscosity coefficients vanish, that is, $\eta = \zeta = \Gamma = 0$, and when the parity-breaking viscosity coefficients satisfy one of the following two conditions*

*Case 1: For $\eta_H(\rho)$ and $\Gamma_H(\rho)$ arbitrary and*

$$G = 0 \, .$$

*Case 2: For an arbitrary function $G(\rho)$ and an arbitrary constant parameter c, along with*

$$\begin{aligned} \eta_H &= cG \, , \\ \Gamma_H &= c(G - 2\rho G') \, . \end{aligned}$$

From Eq. (12), we obtain that $\zeta_H = -\Gamma_H$ in the first case and $\zeta_H = -c(G - 2\rho G') - \rho G'$ in the second one. The conserved energy density in both cases can be generically written as

$$\mathscr{E} = \tfrac{1}{2} \rho v_i^2 + \varepsilon + v_i \partial_i^* G + \frac{c}{\rho} (\partial_i G)^2 \, . \tag{15}$$

Note that when $G = 0$ (Case 1), the energy density has the same functional form as in the inviscid case.

The dissipative nature of viscosities $\eta, \zeta, \Gamma$ is well known. It is not known however that an arbitrary choice of odd viscosities $\eta_H, \zeta_H, \Gamma_H$ may not lead to dissipationless fluid dynamics. The energy density equation can be deduced from Eqs. (1-3) and can be written as

$$\partial_t \left( \varepsilon + \tfrac{1}{2}\rho v_i^2 \right) + \partial_i \left[ \left( \varepsilon' + \tfrac{1}{2}v_j^2 \right) \rho v_i - \eta_{ijkl} v_j \partial_k v_l \right] = -\eta_{ijkl} \partial_i v_j \partial_k v_l \,. \tag{16}$$

In Case 1, the viscosity tensor is given by $\bar{\eta}_{ijkl}$ which forces the right hand side of (16) to vanish due to the antisymmetry property (11), leading to conserved energy density.

The second condition of Statement I is more subtle, and we refer the reader to Appendix A for details. For the particular case of $c = 0$, the only nonvanishing viscosity coefficient is $\zeta_H = -\rho G'$. For this particular case, the corresponding stress is diagonal and can be considered a modification of the pressure term in the Euler equation so that $p \to p - \zeta_H \omega$. Here $\omega = \partial_1 v_2 - \partial_2 v_1$ is the fluid vorticity.

In the next section, we will address when the energy-conserving fluid dynamics described in Statement I can be endowed with the Hamiltonian structure.

## 3 Hamiltonian fluid dynamics in two dimensions

A fluid dynamic system is Hamiltonian if its hydrodynamic equations can be generated by a Hamiltonian function (total energy of the fluid) and a set of Poisson brackets. In other words, both mass and momentum conservation laws can be written as Hamilton's equations. We often refer to the Hamiltonian function together with the Poisson algebra as the Hamiltonian structure. In contrast to the standard textbook examples, here we have both the Hamiltonian, i.e. the integrated energy density of the fluid (Eq. (15)) and the equations of motion (Eqs. (1-3)) together with the conditions of Statement I. Our goal is to derive, when it exists, the Poisson algebra for these systems. As a result of our analysis, we show that not all cases in Statement I can possess Hamiltonian structure.

It is worth to note that the Hamiltonian function need not always be the total energy of system, however we do not consider this possibility in this work. We aim to recover the ideal fluid structure, in the limit of vanishing viscosity coefficients. In addition to that, we only consider local deformations of the ideal fluid Poisson algebra here.

### 3.1 Hamiltonian structure of zeroth-order hydrodynamics

Before we proceed to study the cases of Statement I, let us briefly review the well-known Hamiltonian formulation for the inviscid or, more precisely, the zeroth-order hydrodynamics [17–19]. Let us consider the set of Eqs. (1-3) with $\eta_{ijkl} = 0$. Here it is convenient to write the hydrodynamic equations in terms of conserved quantities since they are frame-independent. Indeed, it is straightforward to check that both equations (1) and (2), written in terms of $\rho$ and $j_i \equiv \rho v_i$, are generated by the Hamiltonian

$$H_0 = \int \left[ \frac{j_i^2}{2\rho} + \varepsilon(\rho) \right] d^2 x \,, \tag{17}$$

along with the following Poisson brackets

$$\{\rho(\boldsymbol{x}), \rho(\boldsymbol{y})\} = 0 \,, \tag{18}$$

$$\{\rho(\boldsymbol{x}), j_i(\boldsymbol{y})\} = -\rho(\boldsymbol{y}) \frac{\partial}{\partial y_i} \delta(\boldsymbol{x} - \boldsymbol{y}) \,, \tag{19}$$

$$\{j_i(\boldsymbol{x}), j_k(\boldsymbol{y})\} = \left[ j_k(\boldsymbol{x}) \frac{\partial}{\partial x_i} - j_i(\boldsymbol{y}) \frac{\partial}{\partial y_k} \right] \delta(\boldsymbol{x} - \boldsymbol{y}) \,. \tag{20}$$

The subalgebra defined in Eq. (20) is the diffeomorphism algebra so that the momentum density $j_i$ is the generator of "local translations" (diffeomorphisms). The Lie-Poisson algebra defined in Eqs. (18-20) is a semidirect product algebra which we will refer to simply as Extended Diffeomorphism Algebra (EDA) hereon. The time evolution of any quantity is defined by $\partial_t Q = \{H, Q\}$.[4] In particular, for the time evolution of density and momentum density fields, one proceeds as

$$\partial_t \rho(\boldsymbol{x}) = \{H, \rho(\boldsymbol{x})\} = \int \left[ \frac{\delta H}{\delta \rho(\boldsymbol{y})} \{\rho(\boldsymbol{y}), \rho(\boldsymbol{x})\} + \frac{\delta H}{\delta j_i(\boldsymbol{y})} \{j_i(\boldsymbol{y}), \rho(\boldsymbol{x})\} \right] d^2 y, \qquad (21)$$

$$\partial_t j_i(\boldsymbol{x}) = \{H, j_i(\boldsymbol{x})\} = \int \left[ \frac{\delta H}{\delta \rho(\boldsymbol{y})} \{\rho(\boldsymbol{y}), j_i(\boldsymbol{x})\} + \frac{\delta H}{\delta j_k(\boldsymbol{y})} \{j_k(\boldsymbol{y}), j_i(\boldsymbol{x})\} \right] d^2 y. \qquad (22)$$

It can be checked that substituting Poisson's brackets (18-20) into the above equations one obtains the correct evolution equations for $\rho$ and $j_i = \rho v_i$ equivalent to (1,2).

Using (18-20) one can compute Poisson brackets of any two functionals of $\rho$ and $j_i$. Poisson brackets between two functions (or functionals) of $\rho$ and $j_i$ must satisfy two conditions: (1) antisymmetry

$$\{Q, R\} = -\{R, Q\}, \qquad (23)$$

and (2) Jacobi identity

$$\mathscr{J}\{Q, R, S\} \equiv \{\{Q, R\}, S\} + \{\{S, Q\}, R\} + \{\{R, S\}, Q\}$$
$$= 0. \qquad (24)$$

Here $\mathscr{J}(Q, R, S)$, defined in the first line of (24), is referred as the *Jacobiator* of three functionals $Q, R, S$ of $\rho$ and $j_i$. The Jacobi identity is the statement that the Jacobiator vanishes for any three functionals.[5] It can be checked that (18-20) satisfy antisymmetry condition and Jacobi identity (23,24).

## 3.2 Modification of brackets for the first-order hydrodynamics

Since not all dynamical systems with conserved energy are Hamiltonian systems [20–22], the scope of this section is to find under which conditions the systems defined in the Statement I are Hamiltonian. For that, we must obtain the brackets that, together with the Hamiltonian

$$H = \int \left[ \frac{j_i^2}{2\rho} + \varepsilon + \frac{j_i}{\rho} \partial_i^* G + \frac{c}{\rho} (\partial_i G)^2 \right] d^2 x, \qquad (25)$$

generate Eqs. (1,2), where the stress tensor satisfy both Eqs. (3-5) and the Statement I. The Hamiltonian (25) is obtained by integrating the energy density (15). It is important to point out that this choice of Hamiltonian is not only natural, but it recovers the ideal fluid Hamiltonian in the limit when $\eta_{ijkl} \to 0$.

We find that the continuity equation can be generated by the brackets (18,19), while the bracket (20) must be modified in order to provide us the correct momentum conservation equation. After such deformation, the momentum density algebra becomes

$$\{j_i(\boldsymbol{x}), j_k(\boldsymbol{y})\} = \left[ j_k(\boldsymbol{x}) \frac{\partial}{\partial x_i} - j_i(\boldsymbol{y}) \frac{\partial}{\partial y_k} \right] \delta(\boldsymbol{x} - \boldsymbol{y})$$
$$- \frac{\partial}{\partial x_j} \left[ \bar{\eta}_{jilk}(\rho(\boldsymbol{x})) \frac{\partial}{\partial x_l} \delta(\boldsymbol{x} - \boldsymbol{y}) \right], \qquad (26)$$

---

[4]Here, we follow the notation convention in Ref. [17] The algebra presented here may differ by an overall negative sign from some other references in the literature.

[5]Jacobi identity is associated to the existence of local canonical coordinates.

where $\bar\eta_{ijkl}$ is defined in (10). The antisymmetry property of $\bar\eta_{ijkl}$ (Eq. (11)) together with the identities

$$f(\boldsymbol{x})\,\delta(\boldsymbol{x}-\boldsymbol{y}) = f(\boldsymbol{y})\,\delta(\boldsymbol{x}-\boldsymbol{y}),$$
$$\frac{\partial}{\partial x_m}\delta(\boldsymbol{x}-\boldsymbol{y}) = -\frac{\partial}{\partial y_m}\delta(\boldsymbol{x}-\boldsymbol{y}),$$

guarantees the antisymmetry of the bracket (26). Here, we assume that the bracket deformation is local and recovers the diffeomorphism algebra (20) in the limit of an ideal fluid. The algebra (18,19,26) is sometime called *almost* Poisson brackets [22], since it satisfies the property (23), but not necessarily the Jacobi identity (24).

The direct computation of equations of motion generated by the Hamiltonian (25) with the algebra (18,19,26) provides us the following hydrodynamic equations

$$\partial_t \rho = -\partial_i \mathrm{j}_i \,, \tag{27}$$

$$\partial_t \mathrm{j}_k = -\partial_i \left[ \frac{\mathrm{j}_i \mathrm{j}_k}{\rho} + p\,\delta_{ik} - \eta^H_{iklm}\,\partial_l\!\left(\frac{\mathrm{j}_m}{\rho}\right) \right], \tag{28}$$

with $\eta^H_{ijkl}$ defined by (9,10) with parameters specified in the Statement I. More specifically, in Case 1, both $\eta_H$ and $\Gamma_H$ are independent functions of density in the bracket (26) and the Hamiltonian is obtained by taking $G = 0$ in the Eq. (25). In Case 2, we must substitute $\eta_H = cG$ and $\Gamma_H = c(G - 2\rho G')$ in the bracket (26). It is easy to see that this system of equations is equivalent to Eqs. (1,2), when the stress tensor is given by Eqs. (3-5) and the viscosity tensor satisfies one of the the cases in the Statement I. Because these equations are generated by antisymmetric brackets, the Hamiltonian (25) itself is automatically conserved, since $\partial_t H = \{H, H\} = 0$. In the following, we will check if these *almost* Poisson brackets satisfy the Jacobi identity.

### 3.3 Constraints imposed by Jacobi identity

*Prima facie* one might think that the identification of antisymmetric brackets (26) means that all energy-conserving cases specified by Statement I are Hamiltonian. However, for the system to be Hamiltonian, brackets must also satisfy the Jacobi identity (24). We now present the main condition for which the brackets in Eqs. (18,19,26) satisfy the Jacobi identity (24).

**Statement II.** *The antisymmetric brackets from Eqs. (18,19,26) are Poisson brackets, i.e, satisfy Jacobi identity if and only if*

$$\Gamma_H(\rho) = \eta_H(\rho) - \rho\,\eta'_H(\rho). \tag{29}$$

*When this condition holds, there exists a locally conserved quantity, namely*

$$J_i \equiv \mathrm{j}_i + \partial_i^* \eta_H(\rho), \tag{30}$$

*which satisfies the diffeomorphism algebra.*

To see the origin of the condition (29) let us consider the modified momentum density $J_i$ (30). It is clear that the bracket $\{\rho(\boldsymbol{x}), J_i(\boldsymbol{y})\}$ is not modified and coincides with (19).

Using brackets (18,19,26) it is straightforward to compute

$$\{J_i(\boldsymbol{x}), J_k(\boldsymbol{y})\} =$$
$$\left[ J_k(\boldsymbol{x})\frac{\partial}{\partial x^i} - J_i(\boldsymbol{y})\frac{\partial}{\partial y^k} \right]\delta(\boldsymbol{x}-\boldsymbol{y}) - (\epsilon_{ji}\delta_{lk} - \delta_{ji}\epsilon_{lk})$$
$$\times \frac{\partial}{\partial x^j}\left[ (\Gamma_H - \eta_H + \rho\eta'_H)(\boldsymbol{x})\frac{\partial}{\partial x^l}\delta(\boldsymbol{x}-\boldsymbol{y}) \right]. \tag{31}$$

One immediately notices that the condition (29) annihilates the second term in the right hand side of Eq. (31) and the algebra of brackets of $\rho$ and $J_i$ becomes identical to the original diffeomorphism algebra (18-20) thereby satisfying the Jacobi identity. Throughout the rest of the paper we will use $J_i$ to refer to the diffeomorphism generators.

In Appendix B, we perform more direct computation showing that the condition (29) is also necessary for brackets (18,19,26) to satisfy the Jacobi identity. The key point of that computation is that the Jacobiator (see Eq. (24)) is given by

$$
\begin{aligned}
\mathscr{J}\Big\{ j_i(x), j_k(y), j_m(z) \Big\} = & \\
\Bigg[ \epsilon_{km} \frac{\partial}{\partial x_i} \frac{\partial}{\partial y_l} \frac{\partial}{\partial z_l} &+ \epsilon_{ik} \frac{\partial}{\partial x_l} \frac{\partial}{\partial y_l} \frac{\partial}{\partial z_m} + \epsilon_{mi} \frac{\partial}{\partial x_l} \frac{\partial}{\partial y_k} \frac{\partial}{\partial z_l} \Bigg] \\
\Big[ 2(\eta_H - \rho\,\eta_H' - \Gamma_H) \delta(x-y)\delta(x-z) \Big] .
\end{aligned}
\tag{32}
$$

The Jacobiator $\mathscr{J}$ vanishes only when (29) holds completing the proof of Statement II.

We now discuss the physical picture behind the constraint (29). The constraint can be rewritten as $\Gamma_H/\rho + \rho(\eta_H/\rho)' = 0$. We consider the angular momentum per particle $\ell/\rho = \eta_H/\rho$ and notice that if it is itself $\rho$-independent, the constraint requires $\Gamma_H = 0$. We see that if one compresses the fluid of rotating particles, no intrinsic torque is needed if the angular momentum of each particle does not depend on the particles' density. If $\eta_H(\rho)$ is nonlinear in $\rho$, the compression would require an additional torque applied to each particle. If (29) is satisfied, this torque can be provided by the intrinsic torque $\Gamma_H$ of the fluid. If the condition (29) is not met, additional "constraint forces" are needed rendering the system non-Hamiltonian.

### 3.4 Conditions for Hamiltonian hydrodynamics

For the first-order hydrodynamics defined by Eqs. (1-5) to be Hamiltonian the viscous stress coefficients in Eq. (5) must jointly satisfy Statement I and Statement II. Case 1 of Statement I defines a Hamiltonian system as long as $\Gamma_H = -\zeta_H = \eta_H(\rho) - \rho\,\eta_H'(\rho)$. For the Case 2, the Jacobi identity constraint is incompatible with the energy conservation condition unless $c = 0$. We summarize these findings in the following statement.

**Statement III.** *The hydrodynamics Eqs. (1-5) is Hamiltonian only in the following cases*
*Case 1: For arbitrary $\eta_H(\rho)$, $G = 0$, and*

$$
\Gamma_H(\rho) = -\zeta_H(\rho) = \eta_H(\rho) - \rho\,\eta_H'(\rho) .
$$

*Case 2: For arbitrary $G(\rho)$ and*

$$
\zeta_H(\rho) = -\rho\, G'(\rho) , \quad \Gamma_H(\rho) = \eta_H(\rho) = 0 .
$$

Furthermore, Case 2 itself comes with a corollary.

**Corollary III.1.** *If the momentum density satisfies the diffeomorphism algebra (20) the only allowed viscosity term in the Hamiltonian is the odd pressure term $\zeta_H$.*

In both cases, the hydrodynamic equations are obtained from the Hamiltonian (25) and the Poisson brackets (18,19) and (26), with the viscosity coefficients satisfying one of the conditions in the Statement III. Note that the Hamiltonian function for Case 1 has the same form as the inviscid Hamiltonian (17). In addition to that, we would like to emphasize that the Hamiltonian hydrodynamics corresponding to Case 1 could be equivalently written in terms of the diffeomorphism generator $J_i$ defined in (30). In these new variables, the Poisson algebra becomes the EDA and the equation of motion for the "modified momentum density" $J_i$

posses higher-order derivative terms and no odd viscosity stress (only odd pressure). This was already pointed out in several references, such as [9, 11, 16] for the particular case of $\Gamma_H = 0$. When $\Gamma_H = 0$, the Jacobi identity condition (29) imposes that $\eta_H(\rho) = \nu_o \rho$, where $\nu_o$ is a constant kinematic odd viscosity. In fact, in the Ref. [11], the authors identify the generators of diffeomorphism $J_i$ to the "molecular" center-of-mass momentum density.

For Case 2, the Hamiltonian is different from that of the inviscid case and can be written as

$$H = \int \left[ \frac{j_i^2}{2\rho} + \varepsilon + \frac{j_i}{\rho} \partial_i^* G \right] d^2 x \,. \tag{33}$$

However, the Poisson brackets remain the same as in the inviscid zeroth-order hydrodynamics, i.e. EDA given by (18,19,20). In this case the full stress tensor is explicitly given by

$$T_{ij} = -\left( p + \rho G' \omega \right) \delta_{ij} \,, \tag{34}$$

which proves the Corollary III.1.

## 3.5 Generalized Hamiltonian hydrodynamics

In the previous sections, we showed that the Hamiltonian structure is intimately related to the existence of hydrodynamic variables $\rho$ and $J_i$, which satisfy the EDA. This way, we can easily generalize our results and propose the most general Hamiltonian hydrodynamics within an appropriate counting scheme.

It is not hard to see that the Poisson algebra (18,19,20) is invariant under the scaling

$$j_i \to \alpha j_i \,, \qquad \partial_i \to \alpha \partial_i \,, \qquad \rho \to \rho \,. \tag{35}$$

Hence, the diffeomorphism generators $J_i$, defined in Eq. (30) also scales as $J_i \to \alpha J_i$. This new counting scheme differs from the original derivative expansion of the stress tensor. Under this scaling, first-order hydrodynamic terms, such as viscous terms in the stress tensor, show up in the same order as the following second-order hydrodynamic terms

$$\tau_{ijkl}(\rho) \partial_k \rho \, \partial_l \rho + \sigma_{ijkl}(\rho) \partial_k \partial_l \rho \,.$$

In the following, we refer to them as Madelung terms.[6]

Note that Eq. (35) together with the continuity equation impose that $\partial_t$ must be of order $\alpha^2$. The scaling (35) is similar to the one used for energy conservation in Appendix A and gives us that the energy density of the fluid must scale in the same way as the fluid stress tensor. Thus, within this counting scheme, the most general Hamiltonian dynamics is given by the following simple prescription. Let us first take the most general Hamiltonian of the second order in $\alpha$, that is,

$$H = \int \left[ \frac{J_i^2}{2\rho} + \varepsilon + \frac{J_i}{\rho} \partial_i^* \mathscr{G} + \frac{1}{2\rho} (\partial_i K)^2 \right] d^2 x \,, \tag{36}$$

where $J_i$ is the diffeomorphism generator, $\mathscr{G}$ and $K$ are arbitrary functions of $\rho$. Let us assume conventional EDA brackets (18-20) and generate equations of motion for $\rho$ and $J_i$. The equation for $\rho$ is the standard continuity equation

$$\partial_t \rho + \partial_i J_i = 0 \,, \tag{37}$$

---

[6]These terms generalize the "quantum pressure" arising from the Madelung transformations in Schrödinger equation.

while the one for $J_i$ is

$$\partial_t J_k + \partial_i\left(\frac{J_i J_k}{\rho} - T_{ik}\right) = 0,\tag{38}$$

with

$$T_{ij} = -\frac{1}{\rho}\partial_i K \partial_j K - \left[p + \rho\,\mathscr{G}'\partial_k\left(\frac{J_k^*}{\rho}\right) - K'\Delta K\right]\delta_{ij}.\tag{39}$$

Once again, pressure is given by (4). In this form the only viscous term present is the odd pressure term with $\zeta_H = -\rho\,\mathscr{G}'$ and primes denote derivatives with respect to $\rho$.

Let us now shift the momentum density according to (30), i.e. $J_i = \mathfrak{j}_i + \partial_i^*\eta_H$, where $\eta_H(\rho)$ is an arbitrary function of $\rho$. As a result, we obtain the hydrodynamic system in terms of these new variables,[7] that is,

$$\partial_t\rho + \partial_i\mathfrak{j}_i = 0,\tag{40}$$

$$\partial_t\mathfrak{j}_k + \partial_i\left(\frac{\mathfrak{j}_i\mathfrak{j}_k}{\rho} - T_{ik}^H\right) = 0,\tag{41}$$

where the new stress tensor $T_{ij}^H$ is given by

$$T_{ij}^H = \eta_{ijkl}^H\partial_k\left(\frac{\mathfrak{j}_l}{\rho}\right) - \frac{A}{\rho}\partial_i\rho\,\partial_j\rho - \delta_{ij}\left[p - B(\nabla\rho)^2 - C\Delta\rho\right],\tag{42}$$

with

$$\zeta_H = -\eta_H - \rho\,\mathscr{G}', \qquad \Gamma_H = \eta_H - \rho\,\eta_H',\tag{43}$$

$$A = K'^2 - \eta_H'^2, \quad C = A + \eta_H'(\mathscr{G}' + \eta_H'),\tag{44}$$

$$B = \frac{1}{2}A' + (\mathscr{G}' + \eta_H')\left(\eta_H'' - \frac{\eta_H'}{\rho}\right).\tag{45}$$

We notice that the modified stress tensor's parity-breaking part is generally defined by two independent functions $\eta_H(\rho)$ and $\mathscr{G}(\rho)$. An additional free function $K(\rho)$ contributes to Madelung terms. The expressions (43-45) are the most general relations on parity-odd coefficients compatible with Hamiltonian hydrodynamics. If by some reason one requires that Madelung term vanish one obtains $K = \pm\eta_H$ and $(\mathscr{G}' + \eta_H')\eta_H' = 0$. The latter equation has two solutions $\mathscr{G} = -\eta_H$ or $\eta_H = 0$. These two solutions give Cases 1 and 2 of the Statement III, respectively. It is interesting to note that the odd viscosity term $\eta_H$ appears in this construction not as a parameter of the Hamiltonian (36) but as the parameter of the momentum density shift or equivalently as a modification of Poisson's brackets.

We remark here that it is relatively straightforward to generate all Hamiltonian systems. One can start with a general local form of the Hamiltonian (36), generate equations using (18-20) and then consider redefinitions of hydrodynamic fields (30) preserving the structure of equations of motion. However, this procedure is based on the assumption that there are no non-trivial extensions of the EDA within the order in derivatives used in this work. *A priori* one might have a non-trivial extension of Poisson algebra similar to the central extensions considered in [23]. The authors are not aware of the theorem on the absence of such extensions, and one should consider the computations done in Appendix B as an explicit proof of such a theorem within our counting scheme.

---

[7]There is a certain ambiguity in the form of the stress tensor resulting from the freedom to add to the stress arbitrary divergenceless terms. These additions, however, do not change the equations of motion.

# 4 Energy conservation and nonholonomic constraints

The absence of Hamiltonian structure in energy-conserving systems is one of the prominent features of so-called nonholonomic systems. These systems are described as systems with restrictions on types of motion. Typical examples include systems like rolling balls and rolling wheels as well as skates with rolling constraints and skating constraints, respectively [21, 22, 24]. In these systems, the constraints imposed on velocities are not integrable and, therefore, cannot be reduced to the constraints on the configurational space of the dynamic system. Such constraints are called nonholonomic and are related to the break down of the Jacobi identity in the Hamiltonian framework [22, 25].

In this section, we consider the fluid dynamics described by Case 1 of Statement I, but not satisfying the condition of Statement II. We show that it arises from Hamiltonian fluid dynamics with internal angular momentum degree of freedom subject to a nonholonomic constraint. The constraint pins the internal angular momentum density to the function of the density of the fluid, preserving energy conservation but breaking the Hamiltonian structure.

Let us consider the following Hamiltonian

$$H_\lambda = \int d^2x \left[ \frac{j_i^2}{2\rho} + \varepsilon(\rho) + \lambda \left( \ell + 2\eta_H(\rho) \right)^2 \right]. \tag{46}$$

This Hamiltonian is a functional of hydrodynamic fields $\rho$ and $j_i$ as well as of the new field $\ell$, which corresponds to the internal angular momentum density of the fluid. The numerical constant $\lambda > 0$ couples the internal angular momentum density to a function of the density of the fluid $\eta_H(\rho)$. For large $\lambda$, it is energetically favorable for the system to have $\ell \approx -2\eta_H$.

Let us assume that the fields obey the Lie-Poisson algebra given by the brackets

$$\{\rho(\boldsymbol{x}), \rho(\boldsymbol{y})\} = \{\ell(\boldsymbol{x}), \rho(\boldsymbol{y})\} = \{\ell(\boldsymbol{x}), \ell(\boldsymbol{y})\} = 0, \tag{47}$$

$$\{\rho(\boldsymbol{x}), j_i(\boldsymbol{y})\} = -\rho(\boldsymbol{y}) \frac{\partial}{\partial y^i} \delta(\boldsymbol{x} - \boldsymbol{y}), \tag{48}$$

$$\{\ell(\boldsymbol{x}), j_i(\boldsymbol{y})\} = \left[ 2\Gamma_H(\rho(\boldsymbol{x})) \frac{\partial}{\partial x^i} - \ell(\boldsymbol{y}) \frac{\partial}{\partial y^i} \right] \delta(\boldsymbol{x} - \boldsymbol{y}), \tag{49}$$

$$\{j_i(\boldsymbol{x}), j_k(\boldsymbol{y})\} = \left[ j_k(\boldsymbol{x}) \frac{\partial}{\partial x^i} - j_i(\boldsymbol{y}) \frac{\partial}{\partial y^k} \right] \delta(\boldsymbol{x} - \boldsymbol{y})$$
$$+ (\epsilon_{ik}\delta_{jl} + \delta_{ik}\epsilon_{jl}) \frac{\partial}{\partial x^j} \left[ \frac{\ell(\boldsymbol{x})}{2} \frac{\partial}{\partial x^l} \delta(\boldsymbol{x} - \boldsymbol{y}) \right]$$
$$- (\delta_{ij}\epsilon_{kl} - \epsilon_{ij}\delta_{kl}) \frac{\partial}{\partial x^j} \left[ \Gamma_H(\rho(\boldsymbol{x})) \frac{\partial}{\partial x^l} \delta(\boldsymbol{x} - \boldsymbol{y}) \right]. \tag{50}$$

One can check that these brackets do satisfy the Jacobi identity. In fact, this is true by construction, since the brackets involving $\ell$ were derived from replacing $\eta_H$ by the new variable $-\frac{1}{2}\ell$ in Eqs. (98, 105). Furthermore, we can recover the conventional EDA presented in [9, 11] if we rewrite this algebra in terms of the quantities $J_i, \rho, L$, which are defined by

$$J_i = j_i - \tfrac{1}{2}\partial_i^* \ell, \tag{51}$$

$$L = \ell + M, \tag{52}$$

with

$$\Gamma_H = \tfrac{1}{2}(M - \rho M'). \tag{53}$$

The Hamiltonian (46) with these brackets define a Hamiltonian fluid dynamics, whose

equations of motion are given by

$$\partial_t \rho + \partial_i(\rho v_i) = 0 \,, \tag{54}$$

$$\partial_t \ell + \partial_i(\ell v_i) = -2\Gamma_H \partial_i v_i \,, \tag{55}$$

$$\partial_t \jmath_j + \partial_i\left(\rho v_i v_j + \tilde{p}\,\delta_{ij}\right) = \partial_i\left[\eta^\ell_{ijkl}\partial_k v_l\right], \tag{56}$$

$$\eta^\ell_{ijkl} = -\tfrac{1}{2}\ell\left(\delta_{ik}\epsilon_{jl} + \delta_{jl}\epsilon_{ik}\right) + \Gamma_H\left(\delta_{ij}\epsilon_{kl} - \epsilon_{ij}\delta_{kl}\right), \tag{57}$$

$$\tilde{p} = \rho\varepsilon' - \varepsilon + \lambda(\delta\ell)^2 - 4\lambda(\eta_H - \rho\eta'_H - \Gamma_H)\delta\ell \,. \tag{58}$$

Here we introduced the notation $\delta\ell = \ell + 2\eta_H$ and $v_i = \jmath_i/\rho$. Notice that the form of the viscous tensor (57) is identical to (10) with the replacement $\ell \to -2\eta_H$.

It is important to understand that the dynamical system (54-58) is Hamiltonian for any value of the parameter $\lambda$ as it is generated by the Hamiltonian (46) with the use of Poisson brackets (47-50). It is clear that, at finite energy, the intrinsic angular momentum $\ell$ should follow $-2\eta_H(\rho)$, in the limit $\lambda \to \infty$. However, from (54,55) we obtain that

$$\partial_t(\delta\ell) + \partial_i(\delta\ell\, v_i) = -2(\Gamma_H - \eta_H + \rho\eta'_H)\partial_i v_i \,. \tag{59}$$

If the condition (29) is satisfied, the right hand side of (59) vanishes. In this case one can start with initial conditions $\ell = -2\eta_H(\rho)$ and the dynamics (59) will preserve these conditions at all times. The constraint

$$-\tfrac{1}{2}\ell = \eta_H(\rho) \tag{60}$$

in this case is the first-class constraint [26] and the substitution of $\ell = -2\eta_H(\rho)$ in all brackets, Hamiltonian and equations is consistent and produces the Hamiltonian dynamics of $\rho$ and $\jmath_i$ specified in the Statement III, Case 1.

Let us assume now that (29) does not hold. In this case, imposing the constraint (60) cannot be reduced to just a choice of initial conditions. Choosing initial conditions satisfying $\ell = -2\eta_H$ we find that, for large but finite $\lambda$, $\ell$ will deviate from $-2\eta_H$ in time, due to (59). However, this deviation creates a large pressure term (58) proportional to $\lambda$ which will lead the flow to be incompressible, that is, $\partial_i v_i = 0$. Consequently, the right hand side of Eq. (59) vanishes, making sure that $\ell \approx -2\eta_H$. The limiting solution at $\lambda \to \infty$ will satisfy the constraint (60) at all times, however it constrains the flow to be incompressible. In other words, in the absence of the restriction (29), the time evolution of the constraint (60) gives rise to incompressibility, which is a second-class constraint [26]. The system with both constraints, that is, Eq. (60) together with $\partial_i v_i = 0$, can be written from a Hamiltonian principle by working out the Dirac brackets of the system [26], which will turn out to be non-local. If, on the other hand, we insist in imposing only the constraint (60), without the incompressibility condition, i.e. neglecting Eq. (59), we end up with a **nonholonomic** constraint. This can be directly observed, if we substitute $\ell$ by $-2\eta_H(\rho)$ in the Poisson bracket (49). This replacement is inconsistent to Eq. (48).

In the following, we explore a possible origin of energy-conserving but non-Hamiltonian fluid dynamics as coming from Hamiltonian systems with additional degrees of freedom through dynamical nonholonomic constraints. The problem of realizing nonholonomic constraints has been considered in the context of dynamical systems. The realization of constraints is not always unique and might result in different equations of motion. We refer the reader to the original article [27] and reviews [21, 22]. The discussion here is heuristic and is closely related to the so-called vakonomic mechanics [28].

Let us introduce a relaxation term proportional to $\delta\ell$ on the right hand side of Eq. (55), that is,

$$\partial_t \ell + \partial_i(\ell v_i) = -2\Gamma_H \partial_i v_i - \gamma\lambda(\delta\ell) \,, \tag{61}$$

with $\gamma > 0$. The energy equation acquires a dissipative term

$$\partial_t \mathscr{E} + \partial_i Q_i = -\gamma \lambda^2 (\delta \ell)^2 \,. \tag{62}$$

This regularization procedure allows us to take the limit $\lambda \to \infty$ without imposing the incompressibility condition. Solving Eq. (61) in powers of $1/\lambda$ gives us

$$\ell = -2\eta_H - \frac{2}{\gamma \lambda}(\Gamma_H - \eta_H + \rho \eta'_H)\,\partial_i v_i + O(\lambda^{-2}) \,. \tag{63}$$

Plugging this expression back in the Eqs. (56-58), we obtain the regularized stress tensor

$$T_{ij} = (-p + \zeta \, \partial_k v_k)\delta_{ij} + \bar{\eta}_{ijkl}\partial_k v_l \,, \tag{64}$$

with the bulk viscosity given by

$$\zeta = \frac{8}{\gamma}(\Gamma_H - \eta_H + \rho \eta'_H)^2 \,. \tag{65}$$

The intrinsic angular momentum relaxation introduces a bulk viscosity in the system and spoils the energy conservation. With the dissipative regularization (61) the limit $\lambda \to \infty$ can be taken. The energy equation (62) in this limit becomes

$$\partial_t \mathscr{E} + \partial_i Q_i = -\zeta (\partial_i v_i)^2 \,, \tag{66}$$

where the bulk viscosity is given by Eq. (65). We observe that the limit $\lambda \to \infty$ produced a family of hydrodynamic equations characterized by the parameter $\gamma$ (compare with [27, 28]). Within this family for $\gamma \to 0$, the bulk viscosity becomes infinite and forces the fluid to be incompressible. This way, we recover the Hamiltonian case with second-class constraints, discussed previously. In the opposite limit $\gamma \to \infty$, the bulk viscosity vanishes, and the system conserves energy, even though it cannot be written from a Hamiltonian principle.

## 5 Discussion and conclusions

We considered a space of two-dimensional fluid dynamics with parity-breaking terms in the viscous stress tensor in this work. We started by identifying the subset of energy-conserving fluids within this space (Statement I). Surprisingly, not all parity-odd viscosity coefficients lead to energy conservation in first-order hydrodynamics. For example, for a hydrodynamic system with $\eta_{ijkl} = \Gamma_H \epsilon_{ij}\delta_{kl}$, Eqs. (77-81) give us

$$\partial_t \mathscr{E} + \partial_i Q_i = -\left[\left(\frac{\Gamma_H}{\rho^2}\right)' \Gamma_H (\partial_i \rho)^2 + \frac{\Gamma_H^2}{\rho^2}\Delta\rho\right]\partial_j v_j \,.$$

The right hand side can be either positive or negative, depending on the flow and the density distribution. Hydrodynamic systems which neither conserve energy nor are exclusively dissipative may be realized in active matter systems, where the driving is local. If, however, we insist on having both energy conservation and $(\eta_H, \Gamma_H, \zeta_H)$ being independent functions, we must allow for Madelung terms in the stress tensor, which is the subject of future work. Some of the hydrodynamical systems considered in this work turn out to be energy-conserving but not Hamiltonian. An obstacle for the system to be Hamiltonian is that the brackets generating equations of motion fail to satisfy the Jacobi identity. We found that this condition amounts to (29) defining what we might refer to as Hamiltonian fluids. We also observed that the bracket generating equations of Hamiltonian fluids could always be transformed to

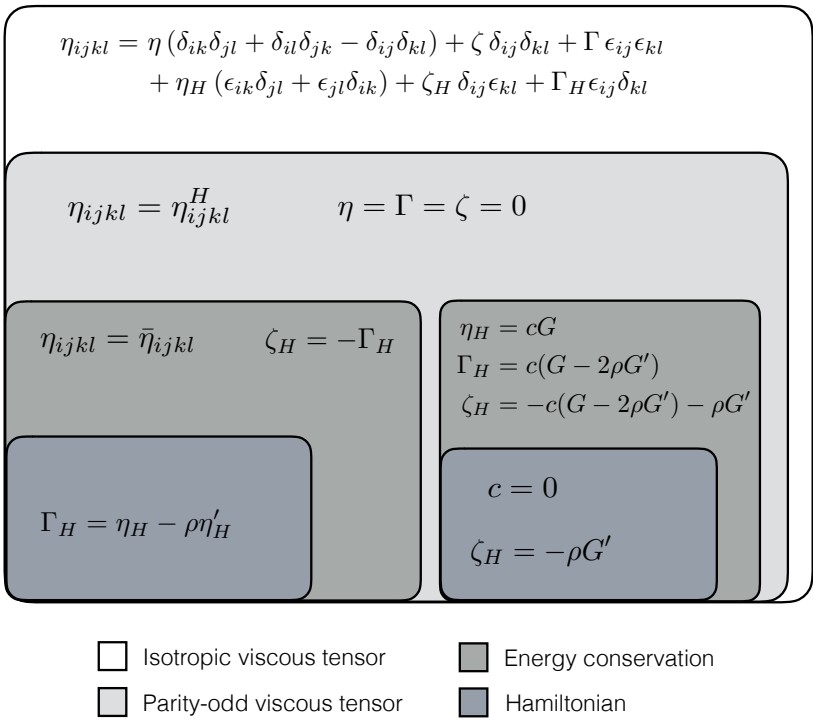

Figure 1: The space of parity breaking barotropic Galilean fluids in two dimensions. In addition to the internal energy density $\varepsilon(\rho)$, the space is parametrized by viscosity coefficients that are considered to be arbitrary functions of density.

the conventional extended diffeomorphism algebra (EDA) (18-20) by changing hydrodynamic variables (Statement II). In particular, if the momentum density satisfies the EDA, the only allowed viscosity term in the Hamiltonian dynamics is the odd pressure $\zeta_H$. The main results supporting the described structure of the space of theories are formulated as Statements I-III with some details of derivations relegated to appendices.

The study of the space of parity-violating hydrodynamic equations in 2+1 dimensions have been done before both in relativistic [13] and nonrelativistic [29,30] context. In this work, we focus on the Hamiltonian fluids. We use a nonrelativistic counting scheme, described in 3.5, to make sure that there are only a finite number of terms in the stress tensor at any given order of the counting scheme. We find the most general Hamiltonian fluid dynamics within the second-order of that counting scheme. This dynamics is characterized by three independent functions of density. The stress tensor (42) and the correspondent transport coefficients are given by Eqs (43-45).

In Section 4, we provided a possible origin of nonholonomic fluid dynamics as originating from fully Hamiltonian extended dynamics with nonholonomic constraints imposed on an additional degree of freedom. This additional degree of freedom, in our case, has a meaning of an intrinsic angular momentum density of the fluid. We introduced an energy cost term in the Hamiltonian (46), such that, in the limit of infinite rigidity ($\lambda \to \infty$), the intrinsic angular momentum ($\ell$) is pinned to the odd viscosity, $\eta_H(\rho)$. Solving for $\ell$, with a particular regularization procedure, we obtain an effective dissipative hydrodynamic system with the stress tensor given by Eqs. (64,65). Therefore, "integrating out" the intrinsic angular momentum density provides us a one-parameter family ($\gamma$) of a dissipative hydrodynamic system. Interestingly enough, we can recover the energy conservation for $\gamma \to 0$ and $\gamma \to \infty$. In the former case, the hydrodynamic system is Hamiltonian and subjected to the incompressibility condition, i.e. $\partial_i v_i = 0$. In the latter, we obtain an energy-conserving system, described in Case 1 of

the Statement I, yet not Hamiltonian, since it does not satisfy the condition of the Statement II.

To conclude, if the stress tensor contains gradient terms, there are both Hamiltonian and energy-conserving nonholonomic fluids. We note that the stability analysis is very different for Hamiltonian and nonholonomic systems [22]. In particular, additional instabilities are expected to occur in nonholonomic systems realizable in active matter.

## Acknowledgements

We want to thank Tom Lubensky, Tomer Markovich, and Boris Khesin for helpful discussions. We specially thank Boris Khesin for bringing Ref. [23] to our attention and Tomer Markovich for carefully reading and suggesting improvements to the manuscript.

**Funding information**   This work is supported by NSF CAREER Grant No. DMR-1944967 (SG) and partly from PSC-CUNY Award. GMM was supported by 21st century foundation startup award from CCNY. This research was supported by grants NSF DMR-1606591 (AGA) and US DOE DESC-0017662 (AGA).

## A   Conditions for energy conservation

Let us consider Eqs. (1-5) and let us work out under which conditions this set of equations allows for a third conserved quantity, namely, energy conservation. One way to do so is to determine the most general form of the energy density and then match all the transport coefficients such that the energy density $\mathcal{E}$ satisfies Eq. (13). To determine the form of $\mathcal{E}$, we need to study the symmetries of Eqs. (1-2). The continuity equation

$$\partial_t \rho + \partial_i(\rho v_i) = 0,$$

is invariant under the following scaling

$$x_i \to x_i/\alpha, \quad t \to t/\beta, \quad \text{and} \quad v_i \to v_i \beta/\alpha. \tag{67}$$

Plugging this scaling into equation

$$\partial_t v_j + v_i \partial_i v_j = \frac{1}{\rho} \partial_i T_{ij} = \frac{1}{\rho} \partial_i \left[ -p(\rho)\delta_{ij} + \eta_{ijkl}(\rho)\partial_k v_l \right],$$

and choosing that $\rho \to \rho$, we obtain

$$T_{ij} \to (\beta/\alpha)^2 T_{ij}. \tag{68}$$

This means that all viscosity coefficients scale as $\beta/\alpha^2$. Since they are only functions of $\rho$, they should have no scaling, which imposes that $\beta = \alpha^2$. Here, one could argue that pressure is also only a function of the density and, thus, should not scale. However, we must note that $p'(\rho) = c_s^2$, where $c_s$ is the sound velocity. Since $c_s$ scales as the velocity flow, we obtain that pressure must scale as $(\beta/\alpha)^2$. The scaling (67) with $\beta = \alpha^2$ fixes the form of energy density. Hence, the most general energy density of order $\alpha^2$, up to total derivatives, is given by

$$\mathcal{E} = \tfrac{1}{2}\rho v_i^2 + \varepsilon(\rho) + F(\rho)\kappa + G(\rho)\omega + \tfrac{1}{2}W(\rho)(\partial_i \rho)^2, \tag{69}$$

where $\omega = \partial_i v_i^*$ is the fluid vorticity, $\kappa = \partial_i v_i$ is the flow compressibility and the functions $F(\rho)$, $G(\rho)$ and $W(\rho)$ must be determined in terms of the viscosity coefficients. *Note that this*

*counting scheme differs from the original derivative expansion and set Madelung terms, such as* $\gamma_{ijkl}(\rho)\partial_k\rho\partial_l\rho + \sigma_{ijkl}(\rho)\partial_k\partial_l\rho$, *to be of the same order as viscous terms.*. For the inviscid case the well-known conserved energy is recovered by setting $F = G = W = 0$.

Taking the time derivative of Eq. (69) give us

$$\partial_t \mathscr{E} = -\partial_i Q_i + F'(\rho)\varepsilon''(\rho)(\partial_i\rho)^2 + A_{ijkl}\,\partial_i v_j \partial_k v_l + B_{ijkl}\,\partial_i\rho\,\partial_j\rho\,\partial_k v_l + C_{ijkl}\,\partial_i\partial_j\rho\,\partial_k v_l\,, \quad (70)$$

where

$$Q_i = \mathscr{E}v_i + T_{ij}v_j - \frac{F\,\delta_{ik} + G\,\epsilon_{ik}}{\rho}\,\partial_j T_{jk} + \rho\kappa W\partial_i\rho - \frac{F'\,\delta_{jm} + G'\,\epsilon_{jm}}{\rho}\,\eta_{imkl}\,\partial_j\rho\,\partial_k v_l\,, \quad (71)$$

$$A_{ijkl} = -\eta_{ijkl} + \frac{F}{2}\epsilon_{ij}\epsilon_{kl} + \frac{F}{2}\left(\delta_{ik}\delta_{jl} + \delta_{il}\delta_{jk} - \delta_{ij}\delta_{kl}\right) + \left(\frac{F}{2} - \rho F'\right)\delta_{ij}\delta_{kl} - \rho\,G'\delta_{ij}\epsilon_{kl}\,, \quad (72)$$

$$B_{ijkl} = \left(\frac{F'}{\rho}\delta_{jm} + \frac{G'}{\rho}\epsilon_{jm}\right)'\eta_{imkl} + \frac{\rho\,W'}{2}\delta_{ij}\delta_{kl} - \frac{W}{2}\left(\delta_{ik}\delta_{jl} + \delta_{il}\delta_{ik} - \delta_{ij}\delta_{kl}\right)\,, \quad (73)$$

$$C_{ijkl} = \left(\frac{F'}{\rho}\delta_{jm} + \frac{G'}{\rho}\epsilon_{jm}\right)\eta_{imkl} + \rho\,W\,\delta_{ij}\delta_{kl}\,. \quad (74)$$

The term $F'\varepsilon''(\partial_i\rho)^2$ is velocity independent and must vanish by itself for the energy to be conserved. This means that either $F'(\rho) = 0$ or $\varepsilon''(\rho) = 0$. However, the sound velocity on a fluid is given by

$$c_s = \sqrt{\rho\varepsilon''(\rho)}\,,$$

which implies that $F'(\rho)$ must necessarily vanish to guarantee energy conservation. Since the energy density is only defined up to total derivatives, we obtain that

$$F(\rho) = 0\,, \quad (75)$$

which substantially simplifies Eqs. (72-74).

The term $A_{ijkl}\,\partial_i v_j\,\partial_k v_l$ is a quadratic form and cannot be written as a total derivative unless $\Gamma = \zeta = -\eta = c_1$. However, as mentioned in the main text, we ignore this particular case, since it does not modify the equations of motion in flat space. Thus, $A_{ijkl}\,\partial_i v_j\,\partial_k v_l$ must necessarily vanish to ensure energy conservation. This is obtained when $A_{ijkl} = -A_{klij}$ and, after imposing Eq. (75), we end up with

$$\eta(\rho) = \zeta(\rho) = \Gamma(\rho) = 0\,, \quad (76)$$
$$\zeta_H(\rho) + \Gamma_H(\rho) + \rho\,G'(\rho) = 0\,. \quad (77)$$

Let us now turn our attention to the last two terms. They give us

$$B_{ijkl}\,\partial_i\rho\,\partial_j\rho\,\partial_k v_l + C_{ijkl}\,\partial_i\partial_j\rho\,\partial_k v_l = \partial_i\left(\frac{\kappa\partial_i\rho - \partial_j\rho\,\partial_j v_i}{\rho}\,2\eta_H G'\right) + \left(\frac{\Gamma_H - \eta_H}{\rho}G' + \rho W\right)\kappa\Delta\rho$$

$$+ \left[\left(\frac{G'}{\rho}\right)'(\Gamma_H - \eta_H) - \frac{\eta_H' G'}{\rho} + \frac{\rho W'}{2}\right]\kappa\,(\partial_i\rho)^2$$

$$+ \left[\frac{\eta_H' G'}{\rho} - \frac{W}{2}\right]\partial_i\rho\,\partial_j\rho\,(\partial_i v_j + \partial_j v_i - \delta_{ij}\kappa)\,. \quad (78)$$

In order to write the right hand side of Eq. (78) as a total derivative, we must impose that

$$\frac{\eta_H' G'}{\rho} - \frac{W}{2} = 0\,, \quad (79)$$

$$\frac{G'}{\rho}(\Gamma_H - \eta_H) + \rho W = 0\,, \quad (80)$$

$$\left(\frac{G'}{\rho}\right)'(\Gamma_H - \eta_H) - \frac{\eta_H' G'}{\rho} + \frac{\rho W'}{2} = 0\,. \quad (81)$$

Equation (77) allows us to express $G(\rho)$ in terms of $\zeta_H(\rho)$ and $\Gamma_H(\rho)$. This means that there are four variables $(\eta_H, \zeta_H, \Gamma_H, W)$ and 3 equations. Unless Eqs. (79-81) are linearly dependent, there is no way to satisfy them for $\eta_H(\rho)$, $\zeta_H(\rho)$ and $\Gamma_H(\rho)$ independent. Plugging Eq. (79) into Eq. (80), we find that

$$\left(\Gamma_H - \eta_H + 2\rho\,\eta_H'\right) G' = 0\,. \tag{82}$$

This breaks into two possible cases, namely, $G'(\rho) = 0$ or $G'(\rho) \neq 0$.

## A.1 Case I: $G'(\rho) = 0$

Let us first consider the case when $G'(\rho) = 0$. Plugging this into Eq. (79) gives us $W = 0$, which is consistent with Eq. (81). From Eq. (77), we see that this case is simply the condition

$$\zeta_H(\rho) = -\Gamma_H(\rho)\,, \tag{83}$$

or equivalently

$$\eta_{ijkl}(\rho) = \bar{\eta}_{ijkl}(\rho)\,, \tag{84}$$

where $\bar{\eta}_{ijkl}$ is defined in Eq. (10).

## A.2 Case II: $G'(\rho) \neq 0$

In this case, Eq. (82) imposes that

$$\Gamma_H(\rho) - \eta_H(\rho) + 2\rho\,\eta_H'(\rho) = 0\,, \tag{85}$$

and Eq. (81) can be written solely in terms of $G'(\rho)$ and $\eta_H(\rho)$. Plugging Eq. (79) into Eq. (81) and expressing $\Gamma_H(\rho)$ in term of $\eta_H(\rho)$ gives us

$$\eta_H'(\rho)\,G''(\rho) - \eta_H''(\rho)\,G'(\rho) = 0\,. \tag{86}$$

Note that $G'(\rho) \neq 0$, otherwise we recover the case I. Therefore, we can express $\eta_H'(\rho)$ in terms of $G'(\rho)$. This gives us

$$\eta_H'(\rho) = c\,G'(\rho)\,, \tag{87}$$

for a constant $c$. Hence, we obtain

$$\eta_H(\rho) = c\,G(\rho) + c_2\,, \tag{88}$$

$$\Gamma_H(\rho) = c\left[G(\rho) - 2\rho\,G'(\rho)\right] + c_2\,, \tag{89}$$

$$\zeta_H(\rho) = -c\,G(\rho) + (2c-1)\rho\,G'(\rho) - c_2\,, \tag{90}$$

$$W(\rho) = \frac{2c}{\rho}\left(G'(\rho)\right)^2 \tag{91}$$

for a generic function $G(\rho)$ and some constant $c_2$. However, if we focus on the stress tensor, we see that

$$\begin{aligned}
T_{ij} = &-\left[p(\rho) + (2c-1)\rho\,G'(\rho)\,\omega\right]\delta_{ij} - 2c\rho\,G'(\rho)\,\kappa\,\epsilon_{ij} \\
&+ \left[c\,G(\rho) + c_2\right]\epsilon_{ik}\,\partial_k v_j\,.
\end{aligned} \tag{92}$$

The constant $c_2$ in the last term does not contribute to equations of motion and we can set it to zero without loss of generality. Moreover, when $c = 0$, only odd pressure is present and the stress tensor becomes diagonal.

## B  Condition to satisfy the Jacobi identity

For the system to be Hamiltonian, the algebra defined through expressions (18), (19), and (26) must satisfy the Jacobi identity. Let us define

$$F_A = \int \left( f_A \rho + \jmath_i \, \xi_A^i \right) d^2 x \,, \tag{93}$$

for some test functions $f_A$, $\xi_A^1$ and $\xi_B^2$. In this notation, Jacobi identity can be written as

$$\epsilon^{ABC} \{\{F_A, F_B\}, F_C\} = 0 \,. \tag{94}$$

Using equation (18), we find that the brackets between $F_A$ and $F_B$ is given by

$$\{F_A, F_B\} = \iint d^2 x \, d^2 y \Big[ \left( f_A(\boldsymbol{x}) \xi_B^i(\boldsymbol{y}) - f_B(\boldsymbol{x}) \xi_A^i(\boldsymbol{y}) \right) \{\rho(\boldsymbol{x}), \jmath_i(\boldsymbol{y})\}$$

$$+ \xi_A^i(\boldsymbol{x}) \xi_B^k(\boldsymbol{y}) \{\jmath_i(\boldsymbol{x}), \jmath_k(\boldsymbol{y})\} \Big],$$

$$\{F_A, F_B\} = \int d^2 x \Big[ \rho \left( \xi_A^i \partial_i f_B - \xi_B^i \partial_i f_A \right) + \jmath_i \left( \xi_A^k \partial_k \xi_B^i - \xi_B^k \partial_k \xi_A^i \right) + \bar{\eta}_{ji\ell k} \, \partial^j \xi_A^i \partial^\ell \xi_B^k \Big]. \tag{95}$$

Plugging the expression (95) into equation (94), we find

$$\epsilon^{ABC} \{\{F_A, F_B\}, F_C\} = \epsilon^{ABC} \iint d^2 x \, d^2 y \Bigg[ 2 \left( \xi_C^i(\boldsymbol{y}) \xi_A^k(\boldsymbol{x}) \frac{\partial f_B}{\partial x^k}(\boldsymbol{x}) - f_C(\boldsymbol{x}) \xi_A^k(\boldsymbol{y}) \frac{\partial \xi_B^i}{\partial x^k}(\boldsymbol{x}) \right)$$

$$\times \{\rho(\boldsymbol{x}), \jmath_i(\boldsymbol{y})\} + 2 \xi_C^k(\boldsymbol{x}) \xi_A^\ell(\boldsymbol{y}) \frac{\partial \xi_B^i}{\partial x^\ell}(\boldsymbol{x}) \{\jmath_i(\boldsymbol{x}), \jmath_k(\boldsymbol{y})\}$$

$$+ f_C(\boldsymbol{y}) \frac{\partial \xi_A^i}{\partial x_j}(\boldsymbol{x}) \frac{\partial \xi_B^k}{\partial x_\ell}(\boldsymbol{x}) \{\bar{\eta}_{ji\ell k}(\boldsymbol{x}), \rho(\boldsymbol{y})\}$$

$$+ \xi_C^m(\boldsymbol{y}) \frac{\partial \xi_A^i}{\partial x_j}(\boldsymbol{x}) \frac{\partial \xi_B^k}{\partial x_\ell}(\boldsymbol{x}) \{\bar{\eta}_{ji\ell k}(\boldsymbol{x}), \jmath_m(\boldsymbol{y})\} \Bigg]. \tag{96}$$

Note that there are two types of terms in equation (96), i.e. some of them depend on 3 vectors $(\xi_A, \xi_B, \xi_C)$, whereas the others depend on 2 vectors $(\xi_A, \xi_B)$ and one function $f_C$. Since they are independent, each type of term must vanish separately. Let us now focus on terms with 2 vectors $(\xi_A, \xi_B)$ and one function $f_C$. The Jacobi identity imposes that

$$\epsilon^{ABC} \int d^2 x \Big[ 2\rho(\boldsymbol{x}) \xi_A^i(\boldsymbol{x}) \xi_B^k \frac{\partial^2 f_C}{\partial x^i \partial x^k}(\boldsymbol{x})$$

$$- \frac{\partial \xi_A^i}{\partial x_j}(\boldsymbol{x}) \frac{\partial \xi_B^k}{\partial x_\ell}(\boldsymbol{x}) \int d^2 y \, f_C(\boldsymbol{y}) \{\bar{\eta}_{ji\ell k}(\boldsymbol{x}), \rho(\boldsymbol{y})\} \Big] = 0 \,,$$

$$\epsilon^{ABC} \iint d^2 x \, d^2 y \frac{\partial \xi_A^i}{\partial x_j}(\boldsymbol{x}) \frac{\partial \xi_B^k}{\partial x_\ell}(\boldsymbol{x}) f_C(\boldsymbol{y}) \{\bar{\eta}_{ji\ell k}(\boldsymbol{x}), \rho(\boldsymbol{y})\} = 0 \,, \tag{97}$$

where in the second line we used that $\epsilon^{ABC} \xi_A^i \xi_B^k$ is antisymmetric in the indices $(i, k)$. Equation (97) imposes that

$$\{\bar{\eta}_{ji\ell k}(\boldsymbol{x}), \rho(\boldsymbol{y})\} = 0 \,, \tag{98}$$

which is automatically satisfied when the components $\bar{\eta}_{ijk\ell}$ are functions solely of $\rho$.

Let us now turn our attention to terms in equation (96) with 3 vectors $(\xi_A, \xi_B, \xi_C)$,

$$\epsilon^{ABC}\int d^2x \left[ 2\,\bar{\eta}_{ji\ell k}(\boldsymbol{x})\frac{\partial \xi_C^k}{\partial x_\ell}(\boldsymbol{x})\frac{\partial}{\partial x_j}\left(\xi_A^m \frac{\partial \xi_B^i}{\partial x^m}\right)(\boldsymbol{x}) - 2 \mathrm{j}_i(\boldsymbol{x})\xi_A^k(\boldsymbol{x})\xi_B^j(\boldsymbol{x})\frac{\partial^2 \xi_C^i}{\partial x^k \partial x^j}(\boldsymbol{x}) \right.$$
$$\left. + \int d^2y\, \xi_C^m(\boldsymbol{y})\frac{\partial \xi_A^i}{\partial x_j}(\boldsymbol{x})\frac{\partial \xi_B^k}{\partial x_\ell}(\boldsymbol{x})\{\bar{\eta}_{ji\ell k}(\boldsymbol{x}), \mathrm{j}_m(\boldsymbol{y})\} \right] = 0,$$

$$\epsilon^{ABC}\int d^2x \left[ \bar{\eta}_{ji\ell k}(\boldsymbol{x})\left( \xi_A^m(\boldsymbol{x})\frac{\partial}{\partial x^m}\left(\frac{\partial \xi_B^i}{\partial x_j}\frac{\partial \xi_C^k}{\partial x_\ell}\right)(\boldsymbol{x}) + 2\frac{\partial \xi_A^m}{\partial x_j}(\boldsymbol{x})\frac{\partial \xi_B^i}{\partial x^m}(\boldsymbol{x})\frac{\partial \xi_C^k}{\partial x_\ell}(\boldsymbol{x}) \right) \right.$$
$$\left. + \int d^2y\, \xi_C^m(\boldsymbol{y})\frac{\partial \xi_A^i}{\partial x_j}(\boldsymbol{x})\frac{\partial \xi_B^k}{\partial x_\ell}(\boldsymbol{x})\{\bar{\eta}_{ji\ell k}(\boldsymbol{x}), \mathrm{j}_m(\boldsymbol{y})\} \right] = 0. \tag{99}$$

In the third line, we used one more time that $\epsilon^{ABC}\xi_A^k \xi_B^j$ is antisymmetric in the indices $(k, j)$ and that

$$\epsilon^{ABC}\bar{\eta}_{ji\ell k}\partial_m\left(\partial^j \xi_B^i \partial^\ell \xi_C^k\right) = \epsilon^{ABC}(\bar{\eta}_{ji\ell k} - \bar{\eta}_{ijk\ell})\partial^\ell \xi_C^k\,\partial_m \partial^j \xi_B^i = 2\epsilon^{ABC}\bar{\eta}_{ji\ell k}\partial^\ell \xi_C^k\,\partial_m \partial^j \xi_B^i.$$

Integrating equation (99) by parts give us

$$\epsilon^{ABC}\int d^2x \left[ 2\bar{\eta}_{ji\ell k}(\boldsymbol{x})\frac{\partial \xi_A^m}{\partial x_j}(\boldsymbol{x})\frac{\partial \xi_B^i}{\partial x^m}(\boldsymbol{x})\frac{\partial \xi_C^k}{\partial x_\ell}(\boldsymbol{x}) - \frac{\partial \xi_B^i}{\partial x_j}(\boldsymbol{x})\frac{\partial \xi_C^k}{\partial x_\ell}(\boldsymbol{x})\frac{\partial}{\partial x^m}\left(\xi_A^m \bar{\eta}_{ji\ell k}\right)(\boldsymbol{x}) \right.$$
$$\left. + \int d^2y\, \xi_A^m(\boldsymbol{y})\frac{\partial \xi_B^i}{\partial x_j}(\boldsymbol{x})\frac{\partial \xi_C^k}{\partial x_\ell}(\boldsymbol{x})\{\bar{\eta}_{ji\ell k}(\boldsymbol{x}), \mathrm{j}_m(\boldsymbol{y})\} \right] = 0. \tag{100}$$

Note that equation (100) is valid for any spatial dimensions, since we still have not used the 2-dimensional form of $\bar{\eta}_{ijk\ell}$. Moreover, if the stress tensor is symmetric, $\bar{\eta}_{ijk\ell} = \bar{\eta}_{jik\ell}$ and the first term vanishes identically. Let us now focus on the first term in (100). Hence,

$$2\epsilon^{ABC}\bar{\eta}_{ji\ell k}\partial^j \xi_A^m \partial_m \xi_B^i \partial^\ell \xi_C^k = \epsilon^{ABC}\bar{\eta}_{ji\ell k}\epsilon^{ji}\epsilon_{nr}\partial^n \xi_A^m \partial_m \xi_B^r \partial^\ell \xi_C^k = 2\Gamma_H \epsilon_{jk}\delta_{i\ell}\partial^j \xi_A^i \partial^\ell \xi_B^k \partial_m \xi_C^m,$$

and equation (100) becomes

$$\epsilon^{ABC}\int d^2x \frac{\partial \xi_B^i}{\partial x_j}(\boldsymbol{x})\frac{\partial \xi_C^k}{\partial x_\ell}(\boldsymbol{x})\left[ \left(\Gamma_H(\boldsymbol{x})(\epsilon_{jk}\delta_{i\ell} + \delta_{jk}\epsilon_{i\ell}) - \bar{\eta}_{ji\ell k}(\boldsymbol{x})\right)\frac{\partial \xi_A^m}{\partial x^m}(\boldsymbol{x}) - \xi_A^m(\boldsymbol{x})\frac{\partial \bar{\eta}_{ji\ell k}}{\partial x^m}(\boldsymbol{x}) \right.$$
$$\left. + \int d^2y\, \xi_A^m(\boldsymbol{y})\{\bar{\eta}_{ji\ell k}(\boldsymbol{x}), \mathrm{j}_m(\boldsymbol{y})\} \right] = 0. \tag{101}$$

However, using that $\epsilon^{ABC}\partial_i \xi_A^i \partial_j \xi_B^j = 0$, together with

$$\epsilon_{jk}\delta_{i\ell} + \delta_{jk}\epsilon_{i\ell} = \epsilon_{j\ell}\delta_{ik} + \delta_{j\ell}\epsilon_{ik},$$

we obtain

$$\epsilon^{ABC}\int d^2x \frac{\partial \xi_B^i}{\partial x_j}(\boldsymbol{x})\frac{\partial \xi_C^k}{\partial x_\ell}(\boldsymbol{x})\left[ \left(\epsilon_{j\ell}\delta_{ik} + \delta_{j\ell}\epsilon_{ik}\right)\left(\left(\Gamma_H(\boldsymbol{x}) - \eta_H(\boldsymbol{x})\right)\frac{\partial \xi_A^m}{\partial x^m}(\boldsymbol{x}) - \xi_A^m(\boldsymbol{x})\frac{\partial \eta_H}{\partial x^m}(\boldsymbol{x})\right) \right.$$
$$\left. - \left(\epsilon_{ji}\delta_{\ell k} - \delta_{ji}\epsilon_{\ell k}\right)\xi_A^m(\boldsymbol{x})\frac{\partial \Gamma_H}{\partial x^m}(\boldsymbol{x}) + \int d^2y\, \xi_A^m(\boldsymbol{y})\{\bar{\eta}_{ji\ell k}(\boldsymbol{x}), \mathrm{j}_m(\boldsymbol{y})\} \right] = 0. \tag{102}$$

Here is convenient to use that $\Gamma_H$ is a function of $\rho$, that is,

$$\{\Gamma_H(\rho(\boldsymbol{x})), \mathrm{j}_m(\boldsymbol{y})\} = -\Gamma_H'(\rho(\boldsymbol{x}))\rho(\boldsymbol{y})\frac{\partial}{\partial y^m}\delta(\boldsymbol{x} - \boldsymbol{y}). \tag{103}$$

Plugging equation (103) into (102), we see that

$$
\epsilon^{ABC} \int d^2x \left[ (\Gamma_H - \eta_H) \frac{\partial \xi_A^m}{\partial x^m} - \xi_A^m \frac{\partial \eta_H}{\partial x^m} + \int d^2y\, \xi_A^m(\boldsymbol{y})\{\bar{\eta}_H(\boldsymbol{x}), \mathsf{j}_m(\boldsymbol{y})\} \right]
$$
$$
\times (\epsilon_{j\ell}\delta_{ik} + \delta_{j\ell}\epsilon_{ik}) \frac{\partial \xi_B^i}{\partial x_j} \frac{\partial \xi_C^k}{\partial x_\ell} = 0 \,.
\tag{104}
$$

The bracket between the odd viscosity and the momentum density is fully determined by Jacobi identity, i.e.

$$
\{\eta_H(\boldsymbol{x}), \mathsf{j}_m(\boldsymbol{y})\} = \left[ \left( \eta_H(\boldsymbol{x}) - \Gamma_H(\rho(\boldsymbol{x})) \right) \frac{\partial}{\partial x^m} + \frac{\partial \eta_H}{\partial x^m}(\boldsymbol{x}) \right] \delta(\boldsymbol{x} - \boldsymbol{y}),
$$
$$
\{\eta_H(\boldsymbol{x}), \mathsf{j}_m(\boldsymbol{y})\} = -\left[ \eta_H(\boldsymbol{y}) \frac{\partial}{\partial y^m} + \Gamma_H(\rho(\boldsymbol{x})) \frac{\partial}{\partial x^m} \right] \delta(\boldsymbol{x} - \boldsymbol{y}).
\tag{105}
$$

So far, we have not used that $\eta_H$ is a function of $\rho$. Imposing it into equation (104) give us

$$
\epsilon^{ABC}(\epsilon_{j\ell}\delta_{ik} + \delta_{j\ell}\epsilon_{ik}) \int d^2x \left[ \Gamma_H(\rho) - \eta_H(\rho) + \rho\, \eta_H'(\rho) \right] \frac{\partial \xi_A^m}{\partial x^m} \frac{\partial \xi_B^i}{\partial x_j} \frac{\partial \xi_C^k}{\partial x_\ell} = 0,
\tag{106}
$$

in other words, the Jacobi identity is only satisfied when

$$
\Gamma_H(\rho) = \eta_H(\rho) - \rho\, \eta_H'(\rho).
\tag{107}
$$

In fact, equation (105) must always be valid, even when $\eta_H$ cannot be expressed in terms of $\rho$. Therefore, equation (105) defines the brackets between the fluid intrinsic angular momentum and momentum density.

Equation (105) however has a deeper implication. Note that

$$
\frac{\partial}{\partial x_j} \left[ \bar{\eta}_{ji\ell k}(\boldsymbol{x}) \frac{\partial}{\partial x_\ell} \right] \delta(\boldsymbol{x} - \boldsymbol{y})
$$
$$
= \frac{\partial}{\partial x_j} \left[ \left( \eta_H(\boldsymbol{x})(\epsilon_{j\ell}\delta_{ik} + \epsilon_{ik}\delta_{j\ell}) + \Gamma_H(\boldsymbol{x})(\epsilon_{ji}\delta_{\ell k} - \delta_{ji}\epsilon_{\ell k}) \right) \frac{\partial}{\partial x_\ell} \right] \delta(\boldsymbol{x} - \boldsymbol{y}),
$$
$$
\frac{\partial}{\partial x_j} \left[ \bar{\eta}_{ji\ell k}(\boldsymbol{x}) \frac{\partial}{\partial x_\ell} \right] \delta(\boldsymbol{x} - \boldsymbol{y}) = \left[ \epsilon_{ij} \frac{\partial}{\partial y_j} \left( \eta_H(\boldsymbol{y}) \frac{\partial}{\partial y^k} \right) - \epsilon_{kj} \frac{\partial}{\partial x_j} \left( \eta_H(\boldsymbol{x}) \frac{\partial}{\partial x^i} \right) \right] \delta(\boldsymbol{x} - \boldsymbol{y})
$$
$$
- \epsilon_{ij} \frac{\partial}{\partial x_j} \left[ \Gamma_H(\boldsymbol{x}) \frac{\partial}{\partial x^k} \right] \delta(\boldsymbol{x} - \boldsymbol{y}) + \epsilon_{kj} \frac{\partial}{\partial y_j} \left[ \Gamma_H(\boldsymbol{x}) \frac{\partial}{\partial y^i} \right] \delta(\boldsymbol{x} - \boldsymbol{y}).
\tag{108}
$$

Using equation (105), we can eliminate the dependence in $\Gamma_H$, since

$$
-\epsilon_{ij} \frac{\partial}{\partial x_j} \left[ \Gamma_H(\boldsymbol{x}) \frac{\partial}{\partial x^k} \right] \delta(\boldsymbol{x} - \boldsymbol{y}) = \{\partial_i^* \eta_H(\boldsymbol{x}), \mathsf{j}_k(\boldsymbol{y})\} - \eta_H(\boldsymbol{y}) \epsilon_{ij} \frac{\partial^2}{\partial y_j \partial y^k} \delta(\boldsymbol{x} - \boldsymbol{y}), \tag{109}
$$
$$
\epsilon_{kj} \frac{\partial}{\partial y_j} \left[ \Gamma_H(\boldsymbol{x}) \frac{\partial}{\partial y^i} \right] \delta(\boldsymbol{x} - \boldsymbol{y}) = \{\mathsf{j}_i(\boldsymbol{x}), \partial_k^* \eta_H(\boldsymbol{y})\} + \eta_H(\boldsymbol{x}) \epsilon_{kj} \frac{\partial^2}{\partial x_j \partial x^i} \delta(\boldsymbol{x} - \boldsymbol{y}). \tag{110}
$$

Therefore, we obtain

$$
\frac{\partial}{\partial x_j} \left[ \bar{\eta}_{ji\ell k}(\boldsymbol{x}) \frac{\partial}{\partial x_\ell} \right] \delta(\boldsymbol{x} - \boldsymbol{y}) = \{\partial_i^* \eta_H(\boldsymbol{x}), \mathsf{j}_k(\boldsymbol{y})\} + \{\mathsf{j}_i(\boldsymbol{x}), \partial_k^* \eta_H(\boldsymbol{y})\}
$$
$$
+ \left[ \partial_i \eta_H(\boldsymbol{y}) \frac{\partial}{\partial y^k} - \partial_k^* \eta_H(\boldsymbol{x}) \frac{\partial}{\partial x^k} \right] \delta(\boldsymbol{x} - \boldsymbol{y}).
\tag{111}
$$

Combining this with equation (26), we see that the quantities $\mathsf{j}_i + \partial_i^* \eta_H$ are the diffeomorphism generators, that is, they satisfy following algebra

$$
\{(\mathsf{j}_i + \partial_i^* \eta_H)(\boldsymbol{x}), (\mathsf{j}_k + \partial_k^* \eta_H)(\boldsymbol{y})\} = \left[ (\mathsf{j}_i + \partial_i^* \eta_H)(\boldsymbol{y}) \frac{\partial}{\partial y^k} - (\mathsf{j}_k + \partial_k^* \eta_H)(\boldsymbol{x}) \frac{\partial}{\partial x^i} \right] \delta(\boldsymbol{x} - \boldsymbol{y}). \tag{112}
$$

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
