# Peer review of "Hamiltonian structure of 2D fluid dynamics with broken parity"

_SciPost Physics, doi:SciPost Phys. 14, 103 (2023)_

## Round 2 · Referee Report · Anonymous (Referee 1) · 2021-11-23

Report

The authors identify new conditions on odd parity transport coefficients in 2D for the resulting hydrodynamic equations to be (i) energy conserving (ii) Hamiltonian. The results are very interesting and should certainly be published in Scipost. There were a few points that I would like the authors to clarify before publication:

1) In determining when these systems are Hamiltonian (the criterion for energy conservation is reasonably unambiguous), the authors assume that (i) the Hamiltonian is the energy (ii) the brackets take the specific form defined by Eqs. (18), (19), (26). Within these assumptions, they derive conditions on the transport coefficients for the system to be Hamiltonian. As far as I can tell, these are sufficient conditions – can the authors rule out the possibility of a different Hamiltonian (i.e. not necessarily the energy) and different Poisson brackets whenever the conditions of Statement III are not met? Or does the authors’ physical interpretation of Statement II show that it captures all possible Hamiltonian cases? As things are written, the claim seems to be a bit vague, e.g. on p3, “we will address when the energy conserving fluid dynamics described in Statement I can be endowed with the Hamiltonian structure”, “we show that not all cases in Statement I can possess Hamiltonian structure”, and Statement III makes no reference to a particular choice of bracket.

2) The authors mention in the abstract that their results constrain odd-parity hydrodynamics that derives from coarse-grained microscopic systems. It seems to me that this is only true if Eqs. (18), (19), (26) also derive from coarse-graining the microscopic canonical brackets in some way – can this be justified? It seems plausible for Eqs. (18) and (19) but less so for Eq. (26). If not, there is the risk that being Hamiltonian is a formal property of the macroscopic hydrodynamics that is unrelated to the microscopic dynamics.

3) Is there any physical significance to the fact that the “counting scheme” on p6 differs from the usual hydrodynamic derivative expansion? Naively this mixing of different orders in the derivative expansion seems difficult to reconcile with hydrodynamic scaling (e.g. the “Madelung terms” are dispersive and usually neglected in low-order hydrodynamics)

4) The explicit connection to nonholonomic constraints in Sec. IV is nice. Is this expected in all cases that violate Statement III? (there seems to be a claim in the literature that violating Jacobi is always equivalent to a nonholonomic constraint in finite dimensions, e.g. Van Der Schaft, Maschke, Rep. Math. Phys. 34 2 p.225-233, 1994). The way things are currently stated in Sec. IV seems possibly overly general: “In this section, we consider the fluid dynamics described by Case 1 of Statement I, but not satisfying the condition of Statement II. We show that it arises from Hamiltonian fluid dynamics with internal angular momentum degree of freedom subject to a nonholonomic constraint”, given that it is illustrated with one specific example.

5) The conclusion felt a little technical and confusing, with references back to very specific details and some new details added (e.g. central extensions). Please consider revising.

Finally, notation/typos:
1) I found v_i^2 in Eqs. 14-16 a bit confusing, similarly j_i^2 elsewhere. Could the authors consider an alternative notation?
2) Appendices: what (four times) -> which
  • validity: -
  • significance: -
  • originality: -
  • clarity: -
  • formatting: -
  • grammar: -

Author:  Gustavo Machado Monteiro  on 2022-01-25  [id 2120]

(in reply to Report 1 on 2021-11-23)

The authors identify new conditions on odd parity transport coefficients in 2D for the resulting hydrodynamic equations to be (i) energy conserving (ii) Hamiltonian. The results are very interesting and should certainly be published in Scipost. There were a few points that I would like the authors to clarify before publication:

Authors: We would like to thank the referee for carefully going over the manuscript and recommending it for publication. The referee raised useful questions that helped us to improve the manuscript. Below are the responses to the specific questions raised by the referee.

1) In determining when these systems are Hamiltonian (the criterion for energy conservation is reasonably unambiguous), the authors assume that (i) the Hamiltonian is the energy (ii) the brackets take the specific form defined by Eqs. (18), (19), (26). Within these assumptions, they derive conditions on the transport coefficients for the system to be Hamiltonian. As far as I can tell, these are sufficient conditions – can the authors rule out the possibility of a different Hamiltonian (i.e. not necessarily the energy) and different Poisson brackets whenever the conditions of Statement III are not met? Or does the authors’ physical interpretation of Statement II show that it captures all possible Hamiltonian cases? As things are written, the claim seems to be a bit vague, e.g. on p3, “we will address when the energy conserving fluid dynamics described in Statement I can be endowed with the Hamiltonian structure”, “we show that not all cases in Statement I can possess Hamiltonian structure”, and Statement III makes no reference to a particular choice of bracket.

Authors: Although we have not explicitly worked out the necessary conditions for such, in the manuscript, we considered that:

I. We must recover the ideal fluid Hamiltonian and Poisson algebra in the limit when the viscosity tensor vanishes;

II. The algebra must be local (necessary for uniqueness of the brackets).

These are fairly restrictive and one can believe our results are not only sufficient, but necessary conditions. In fact, assumption I states that to obtain a Hamiltonian system with a different Hamiltonian, we first need to show that there is another conserved quantity that recovers the ideal fluid Hamiltonian in the limit when \eta_{ijkl}\rightarrow0.

We have included these 2 assumptions in the text.

2) The authors mention in the abstract that their results constrain odd-parity hydrodynamics that derives from coarse-grained microscopic systems. It seems to me that this is only true if Eqs. (18), (19), (26) also derive from coarse-graining the microscopic canonical brackets in some way – can this be justified? It seems plausible for Eqs. (18) and (19) but less so for Eq. (26). If not, there is the risk that being Hamiltonian is a formal property of the macroscopic hydrodynamics that is unrelated to the microscopic dynamics.

Authors: If coarse-graining procedure starts from a Hamiltonian system and this process is done consistently, the resulting system should preserve the Hamiltonian structure. In fact, we show in the Statement II that the hydrodynamic fields can be redefined (see Eq. (30)) in such a way that, starting from Eq. (26), we recover the diffeomorphism algebra, i.e., Eq. (20). From a physical point of view, one can always identify these diffeomorphism generators to the density of the center-of-mass molecule momenta (as it has been done in the Ref. [11]).

The starting point of our work is already a hydrodynamic system, which can be viewed as the end product of a consistent coarse-graining procedure.

3) Is there any physical significance to the fact that the “counting scheme” on p6 differs from the usual hydrodynamic derivative expansion? Naively this mixing of different orders in the derivative expansion seems difficult to reconcile with hydrodynamic scaling (e.g. the “Madelung terms” are dispersive and usually neglected in low-order hydrodynamics)

Authors: In this paper, we consider Eqs. (1-4) as our starting point. From here, one can ask if these equations allow for a third conserved quantity, namely the fluid energy. Since we are interested in Hamiltonian systems, the energy must be fully conserved, without neglecting higher-order derivatives. From this viewpoint, this counting scheme is very natural, as can be seen in the Appendix A. Moreover, this is a natural scheme in the Hamiltonian fluid dynamics, as one can see that preserves the Poisson structure (18-20) and (26).

From a more physical ground, such a counting scheme seems to be more appropriate in the non-relativistic hydrodynamics, which allows the fluctuations of the fluid density to be of the same order of the velocity flow. In fact, we will address this in a future work, namely, using the count scheme to determine the most general conditions on the transport coefficients to ensure the non-positive production of fluid energy. It is also an interesting project to understand this counting scheme from the Chapman-Enskog point of view in kinetic theory.

4) The explicit connection to nonholonomic constraints in Sec. IV is nice. Is this expected in all cases that violate Statement III? (there seems to be a claim in the literature that violating Jacobi is always equivalent to a nonholonomic constraint in finite dimensions, e.g. Van Der Schaft, Maschke, Rep. Math. Phys. 34 2 p.225-233, 1994). The way things are currently stated in Sec. IV seems possibly overly general: “In this section, we consider the fluid dynamics described by Case 1 of Statement I, but not satisfying the condition of Statement II. We show that it arises from Hamiltonian fluid dynamics with internal angular momentum degree of freedom subject to a nonholonomic constraint”, given that it is illustrated with one specific example.

Authors: In fact, the failure of the Jacobi identity is the definition of a nonholonomic system, in the Hamiltonian framework (Ref. [27]). We added a sentence in main text stating that.

We were not aware of such reference when writing the paper. We thank the referee to bring it to our attention, we cited this paper in the revised version.

Regarding whether our approach in the Sec. IV is more general than we presented there or not, we believe so. However, since this was supposed to be just an illustrative example, we have not worked it out for the most general case. We will address this method to a larger class of nonholonomic system in a future publication.

5) The conclusion felt a little technical and confusing, with references back to very specific details and some new details added (e.g. central extensions). Please consider revising.

Authors: We agree with the referee on this regard and we removed some of the more technical discussion.

Finally, notation/typos: I found v_i^2 in Eqs. 14-16 a bit confusing, similarly j_i^2 elsewhere. Could the authors consider an alternative notation?

Authors: To the best of our knowledge, this is a fairly standard notation. In this revised version, we did however explain it when this first appears.

2) Appendices: what (four times) -> which

Authors: Fixed.

GMM, AGA and SG.

---

## Round 2 · Referee Report · Anonymous (Referee 2) · 2022-2-18

Report

The paper considers non-relativistic parity violating fluids in 2 spatial dimensions, and asks, whether the non-dissipative part of the system can be described by Hamiltonian dynamics. In general, a non-relativistic fluid is described by a stress-energy-momentum complex. The authors parameterize the stress tensor in terms of viscosity coefficients, and constrain them by Galilean invariance. The question then is when the dynamical system has a conserved energy operator. The authors address this by carrying out a classical phase space analysis. The discussion is interesting and helps understand some of the structure of parity-odd terms in 2 spatial dimensions. The paper is also clearly written. I recommend it be accepted for publication.

---

## Round 3 · Referee Report · Anonymous (Referee 1) · 2022-7-12

Report
p3: “For a hydrodynamic system whose energy is conserved, we derive constraints on the transport coefficients for the system to be Hamiltonian. As a consequence, we obtain that an energy-conserving hydrodynamic system is Hamiltonian only if there exists a conserved quantity, ρvi + εi j∂jηH , which satisfies the diffeomorphism algebra”
I believe that the authors have derived “sufficient conditions”, and not “constraints”. Similarly the “only if” here should be replaced by an “if”.
p6. “In the next section, we will address when the energy-conserving fluid dynamics described in Statement I can be endowed with the Hamiltonian structure”
I again find that the authors have only addressed sufficient conditions for existence of a Hamiltonian structure.
p6. “A fluid dynamic system is Hamiltonian if its hydrodynamic equations can be generated by a Hamiltonian function (total energy of the fluid) and a set of Poisson bracket”
“In contrast to the standard textbook examples, here we have both the Hamiltonian, i.e., the integrated energy density of the fluid…”
The Hamiltonian in a Hamiltonian structure for PDEs is not always the total energy. Please consider revising to make clear that the identification Hamiltonian=energy is a specific choice being made here.
p6. “Our goal is to derive, when it exists, the Poisson algebra for these systems. As a result of our analysis, we show that not all cases in Statement I can possess Hamiltonian structure.”
It seems to me that the authors have constructed a specific bracket that is only symplectic for a proper subset of the cases covered by Statement I. Without further qualification, this is not the same as showing that some cases cannot possess a Hamiltonian structure.
It would be helpful to make clear already at this stage the specific choice that the “the bracket deformation is local and recovers the diffeomorphism algebra (20) in the limit of an ideal fluid”. Even with these assumptions in place, it is not clear to me why the deformation in Eq. (26) is the most general deformation consistent with the EoM and specific choice of Hamiltonian.
p9-10: “For the first-order hydrodynamics defined by Eqs. (1-5) to be Hamiltonian the viscous stress coefficient…”, “defines a Hamiltonian system”, “is Hamiltonian”
Again I find this use of the word “Hamiltonian” confusing, given that the authors are making very specific choices of Hamiltonian function and brackets. I recommend putting these choices explicitly in Statement III, rather than below Statement III as they are currently.
There were several other somewhat ambiguous uses of the word “Hamiltonian” in the remainder of the manuscript, but I leave to these to the authors’ judgement.
Author: Gustavo Machado Monteiro on 2022-12-20 [id 3165]
(in reply to Report 2 on 2022-07-12)We thank the referee for a comprehensive and thorough review. We were able to improve the manuscript through these exchanges. We have implemented the necessary changes requested by the referee.
We have replaced "constraints" by "sufficient conditions" and "only if" by "if" as suggested by the referee. In addition to that, we addressed the other points by including the following paragraph immediately before the section 3.1:
"It is worth to note that the Hamiltonian function need not always be the total energy of system, however we do not consider this possibility in this work. We aim to recover the ideal fluid structure, in the limit of vanishing viscosity coefficients. In addition to that, we only consider local deformations of the ideal fluid Poisson algebra here. "

---

## Round 3 · Author Response

The authors identify new conditions on odd parity transport coefficients in 2D for the resulting hydrodynamic equations to be (i) energy conserving (ii) Hamiltonian. The results are very interesting and should certainly be published in Scipost. There were a few points that I would like the authors to clarify before publication:
Authors: We would like to thank the referee for carefully going over the manuscript and recommending it for publication. The referee raised useful questions that helped us to improve the manuscript. Below are the responses to the specific questions raised by the referee.
1) In determining when these systems are Hamiltonian (the criterion for energy conservation is reasonably unambiguous), the authors assume that (i) the Hamiltonian is the energy (ii) the brackets take the specific form defined by Eqs. (18), (19), (26). Within these assumptions, they derive conditions on the transport coefficients for the system to be Hamiltonian. As far as I can tell, these are sufficient conditions – can the authors rule out the possibility of a different Hamiltonian (i.e. not necessarily the energy) and different Poisson brackets whenever the conditions of Statement III are not met? Or does the authors’ physical interpretation of Statement II show that it captures all possible Hamiltonian cases? As things are written, the claim seems to be a bit vague, e.g. on p3, “we will address when the energy conserving fluid dynamics described in Statement I can be endowed with the Hamiltonian structure”, “we show that not all cases in Statement I can possess Hamiltonian structure”, and Statement III makes no reference to a particular choice of bracket.
Authors: Although we have not explicitly worked out the necessary conditions for such, in the manuscript, we considered that:
I. We must recover the ideal fluid Hamiltonian and Poisson algebra in the limit when the viscosity tensor vanishes;
II. The algebra must be local (necessary for uniqueness of the brackets).
These are fairly restrictive and one can believe our results are not only sufficient, but necessary conditions. In fact, assumption I states that to obtain a Hamiltonian system with a different Hamiltonian, we first need to show that there is another conserved quantity that recovers the ideal fluid Hamiltonian in the limit when \eta_{ijkl}\rightarrow0.
We have included these 2 assumptions in the text.
2) The authors mention in the abstract that their results constrain odd-parity hydrodynamics that derives from coarse-grained microscopic systems. It seems to me that this is only true if Eqs. (18), (19), (26) also derive from coarse-graining the microscopic canonical brackets in some way – can this be justified? It seems plausible for Eqs. (18) and (19) but less so for Eq. (26). If not, there is the risk that being Hamiltonian is a formal property of the macroscopic hydrodynamics that is unrelated to the microscopic dynamics.
Authors: If coarse-graining procedure starts from a Hamiltonian system and this process is done consistently, the resulting system should preserve the Hamiltonian structure. In fact, we show in the Statement II that the hydrodynamic fields can be redefined (see Eq. (30)) in such a way that, starting from Eq. (26), we recover the diffeomorphism algebra, i.e., Eq. (20). From a physical point of view, one can always identify these diffeomorphism generators to the density of the center-of-mass molecule momenta (as it has been done in the Ref. [11]).
The starting point of our work is already a hydrodynamic system, which can be viewed as the end product of a consistent coarse-graining procedure.
3) Is there any physical significance to the fact that the “counting scheme” on p6 differs from the usual hydrodynamic derivative expansion? Naively this mixing of different orders in the derivative expansion seems difficult to reconcile with hydrodynamic scaling (e.g. the “Madelung terms” are dispersive and usually neglected in low-order hydrodynamics)
Authors: In this paper, we consider Eqs. (1-4) as our starting point. From here, one can ask if these equations allow for a third conserved quantity, namely the fluid energy. Since we are interested in Hamiltonian systems, the energy must be fully conserved, without neglecting higher-order derivatives. From this viewpoint, this counting scheme is very natural, as can be seen in the Appendix A. Moreover, this is a natural scheme in the Hamiltonian fluid dynamics, as one can see that preserves the Poisson structure (18-20) and (26).
From a more physical ground, such a counting scheme seems to be more appropriate in the non-relativistic hydrodynamics, which allows the fluctuations of the fluid density to be of the same order of the velocity flow. In fact, we will address this in a future work, namely, using the count scheme to determine the most general conditions on the transport coefficients to ensure the non-positive production of fluid energy. It is also an interesting project to understand this counting scheme from the Chapman-Enskog point of view in kinetic theory.
4) The explicit connection to nonholonomic constraints in Sec. IV is nice. Is this expected in all cases that violate Statement III? (there seems to be a claim in the literature that violating Jacobi is always equivalent to a nonholonomic constraint in finite dimensions, e.g. Van Der Schaft, Maschke, Rep. Math. Phys. 34 2 p.225-233, 1994). The way things are currently stated in Sec. IV seems possibly overly general: “In this section, we consider the fluid dynamics described by Case 1 of Statement I, but not satisfying the condition of Statement II. We show that it arises from Hamiltonian fluid dynamics with internal angular momentum degree of freedom subject to a nonholonomic constraint”, given that it is illustrated with one specific example.
Authors: In fact, the failure of the Jacobi identity is the definition of a nonholonomic system, in the Hamiltonian framework (Ref. [27]). We added a sentence in main text stating that.
We were not aware of such reference when writing the paper. We thank the referee to bring it to our attention, we cited this paper in the revised version.
Regarding whether our approach in the Sec. IV is more general than we presented there or not, we believe so. However, since this was supposed to be just an illustrative example, we have not worked it out for the most general case. We will address this method to a larger class of nonholonomic system in a future publication.
5) The conclusion felt a little technical and confusing, with references back to very specific details and some new details added (e.g. central extensions). Please consider revising.
Authors: We agree with the referee on this regard and we removed some of the more technical discussion.
Finally, notation/typos: I found v_i^2 in Eqs. 14-16 a bit confusing, similarly j_i^2 elsewhere. Could the authors consider an alternative notation?
Authors: To the best of our knowledge, this is a fairly standard notation. In this revised version, we did however explain it when this first appears.
2) Appendices: what (four times) -> which
Authors: Fixed.
GMM, AGA and SG.

---

## Round 3 · List of Changes

Listed in the author comments/response.

---

## Round 4 · Referee Report · Anonymous (Referee 1) · 2023-1-10

Report

I thank the authors for their clarifications and am happy to recommend the manuscript for publication.

---

## Round 4 · List of Changes

We have replaced “constraints” by “sufficient conditions” and “only if” by “if” in the following paragraph on p. 03:

“For a hydrodynamic system whose energy is conserved, we derive sufficient conditions on the transport coefficients for the system to be Hamiltonian. As a consequence, we obtain that an energy-conserving hydrodynamic system is Hamiltonian if there exists a conserved quantity, ρvi + εi j∂jηH , which satisfies the diffeomorphism algebra”.

In addition to that, we have included the following paragraph on p.06:

"It is worth to note that the Hamiltonian function need not always be the total energy of
system, however we do not consider this possibility in this work. We aim to recover the ideal
fluid structure, in the limit of vanishing viscosity coefficients. In addition to that, we only consider
local deformations of the ideal fluid Poisson algebra here".

---

## Editorial Decision

published